# Orthophosphate increases the efficiency of slow muscle-myosin isoform in the presence of omecamtiv mecarbil

Serena Governali[1,2,4], Marco Caremani [1,4], Cristina Gallart[1], Irene Pertici [1], Ger Stienen[2,3], Gabriella Piazzesi [1], Coen Ottenheijm[2], Vincenzo Lombardi [1✉] & Marco Linari [1]

Omecamtiv mecarbil (OM) is a putative positive inotropic tool for treatment of systolic heart dysfunction, based on the finding that in vivo it increases the ejection fraction and in vitro it prolongs the actin-bond life time of the cardiac and slow-skeletal muscle isoforms of myosin. OM action in situ, however, is still poorly understood as the enhanced $Ca^{2+}$-sensitivity of the myofilaments is at odds with the reduction of force and rate of force development observed at saturating $Ca^{2+}$. Here we show, by combining fast sarcomere-level mechanics and ATPase measurements in single slow demembranated fibres from rabbit soleus, that the depressant effect of OM on the force per attached motor is reversed, without effect on the ATPase rate, by physiological concentrations of inorganic phosphate (Pi) (1-10 mM). This mechanism could underpin an energetically efficient reduction of systolic tension cost in OM-treated patients, whenever [Pi] increases with heart-beat frequency.

[1] PhysioLab, Department of Biology, University of Florence, Sesto Fiorentino 50019, Italy. [2] Department of Physiology, Amsterdam UMC (location VUmc), 1081 HZ, Amsterdam, The Netherlands. [3]Present address: Department of Physiology, Kilimanjaro Christian Medical University College, Moshi, Tanzania. [4]These author contributed equally: Serena Governali, Marco Caremani. ✉email: vincenzo.lombardi@unifi.it

In the heart, mutations in the myosin motor are responsible for cardiomyopathies, which consist either in an increased contractility with reduced ventricular filling or diastolic dysfunction (hypertrophic cardiomyopathy, HCM) or in a decreased contractility with reduced ejection fraction or systolic dysfunction (dilated cardiomyopathy)[1]. A promising approach for systolic dysfunction treatment is the development of small molecules that, acting as cardiac myosin activators, are able to increase the contractility during the systole. Among them, omecamtiv mecarbil (OM) has been found to increase the systolic ejection fraction[2–6] and is currently in phase-three clinical trial[7]. OM acts on both α and β cardiac myosin heavy chain (MHC) isoforms and on the slow-skeletal MHC isoform[8]. Noteworthy, the β cardiac MHC isoform, which accounts for >95% of the myosin in the human ventricle, is encoded by the same gene (*MYH7*) as the slow-skeletal MHC isoform[8–12] and the two isoforms will be named β/slow MHC hereafter. OM binds to the catalytic domain of the myosin molecule between the nucleotide binding pocket and the converter domain, a communicating zone between the active site and the lever arm responsible for the working stroke. As a consequence, the release of inorganic phosphate (Pi) is promoted and the lever arm of the molecular motor is stabilized in a pre-working stroke position[2,13,14]. Accordingly, the velocity of actin sliding in the in vitro motility assay is reduced by OM[15–17]. A recent in vitro mechanical study[9] has significantly advanced the understanding of OM mechanism, showing that OM binding to β-cardiac myosin inhibits the working stroke and prolongs the actin-attachment life time of the motor. In this way, in situ, OM may increase thin filament activation at low $Ca^{2+}$ allowing recruitment of OM-free motors, able to generate force. However, a detailed in situ analysis of the effects of OM on the mechanokinetics and energetic of myosin motors has still to be done.

Here we determine, in $Ca^{2+}$-activated demembranated fibres from rabbit soleus, the effects of OM on either the number and force of the attached myosin motors or their ATPase activity and how physiological concentrations of inorganic phosphate (Pi) influence these parameters. The slow-skeletal muscle of the rabbit has been chosen instead of the heart because their myosin isoforms exhibit similar affinity for OM[8], while the required nanometre-microsecond resolution of sarcomere-level mechanics can only be achieved in demembranated myocytes from skeletal muscle[18,19]. Intact trabeculae dissected from the ventricle of the rat heart, on which sarcomere-level mechanics has been recently successfully exploited[20,21], do not suit the present investigation because (*i*) only skinned myocytes allow the required manipulation of $[Ca^{2+}]$ and [Pi] and (*ii*), only ~20% of the myosin is β/slow MHC isoform and the remaining ~80% is α MHC isoform[21]. Determining the fraction of motors recruited for actin interaction during $Ca^{2+}$-activated isometric contraction of demembranated soleus fibres allowed the demonstration that OM reduces the average force per attached motor but at low $Ca^{2+}$ it increases the number of attached motors. The inhibitory effect of OM on force generation is reversed by increasing [Pi] (range 1–10 mM), which acts as an allosteric competitor that allows the OM-bound no-force-generating motors to release OM and re-enter the force-generating cycle without any change in the rate of ATP hydrolysis. We conclude that in the heart systole, in which the internal $[Ca^{2+}]$ remains below the saturating level, the inotropic action of OM occurs via two mechanisms: the first is the increase in the number of attached motors, the second, emerging at physiological levels of intracellular Pi (1–10 mM[22–28]), is a Pi-dependent recovery of the force per motor. This effect of Pi on the OM-bound motors represents an energetically efficient inotropic response in patients under OM treatment, whenever the increase of heart-beat frequency increases [Pi][27,28].

## Results

**Titration of OM effect on the isometric force development.** Following a temperature jump from 1 to 12 °C in the activating solution at saturating $[Ca^{2+}]$ (pCa 4.5), the force rises from near zero to a steady value ($T_0$) with an almost exponential time course (Fig. 1a). Addition of 1 µM OM halves $T_0$ and slows the force development that shows a sigmoidal shape (Fig. 1b). In eight fibres $T_0$ decreases from $146 \pm 5$ kPa in the control to $64 \pm 3$ kPa in the presence of 1 µM OM. The reciprocal of the half-time for force development, $r_{TD}$, taken as an estimate of the rate of force development, is reduced by 1 µM OM from $3.7 \pm 0.2$ s$^{-1}$ to $1.9 \pm 0.1$ s$^{-1}$. The reduction of both $T_0$ and $r_{TD}$ by OM is dose-dependent as shown by the relations of $T_0$ (Fig. 1c) and $r_{TD}$ (d) versus [OM]. The effect nearly saturates at 10 µM OM when $T_0$ is ~35 kPa and $r_{TD}$ is ~1.4 s$^{-1}$. The concentration of OM for half-maximal effect is ~0.5 µM for both $T_0$ and $r_{TD}$.

**OM does not alter the fraction of motors at full activation.** The half-sarcomere stiffness at $T_0$ ($k_0$) during maximal $Ca^{2+}$ activation is estimated from the slope of the $T_1$ relation determined, both in the control and in the presence of OM (range 0.1–10 µM), by plotting the force attained in response to stepwise changes in fibre length against the corresponding changes in the half-sarcomere length measured with the striation follower (see "Methods") (Fig. 2a, b). The $T_1$ relations are shown in Fig. 2c in both control (filled circles) and 1 µM OM (open circles). Both the ordinate intercept ($T_0$) and the abscissa intercept ($Y_0$, an estimate of the half-sarcomere strain at $T_0$) reduce in OM (open circles) by a similar amount (~60%) relative to the control (filled circles). Thus, the slope of the $T_1$ relation, $k_0$, is the same in the presence of OM as in control. In the eight fibres used for these experiments, $k_0$ is $28.1 \pm 0.7$ kPa nm$^{-1}$ in control and remains almost constant independent of [OM] in the whole range of concentrations used (0.1–10 µM, Fig. 2d, triangles), with a mean value of $29.8 \pm 0.6$ kPa nm$^{-1}$ (dashed line). In the same range of [OM], $Y_0$ linearly depends on the isometric force and the extrapolation of the relation to zero force crosses the origin (Fig. 2e). This result indicates that the reduction of isometric force by OM at saturating $[Ca^{2+}]$ is due to a proportional reduction in the strain of all the elastic elements in the half-sarcomere (myofilaments and attached myosin motors) and thus OM reduces the half-sarcomere force by reducing the average force per attached myosin motor without changing the number of attached motors.

This conclusion is supported quantitatively by extracting the stiffness of the array of attached motors, $e_0$ (Fig. 2f), from $k_0$. $e_0$ is determined by subtracting the contribution of the filament compliance ($C_f$) from the half-sarcomere compliance (see "Methods"). The procedure is based on the definition of the half-sarcomere in terms of a simple mechanical model in which the attached motors, represented by an array of in parallel springs with average strain $s_0$[18], are in series with the actin and myosin filament elasticity represented by a spring with an equivalent filament compliance $C_f$. The values of $C_f$ and $s_0$ are reported in Table 1. Knowing $C_f$ (15.2 nm MPa$^{-1}$, average from Table 1), the dependence of $e_0$ on [OM] can be calculated from the relation between $k_0$ and [OM] (Eq. (1)). As shown in Fig. 2f, $e_0$ remains almost constant (~55 kPa nm$^{-1}$) independent of [OM]. Under the sensible assumption that OM does not affect the stiffness of the motor, $e_0$ is an estimate of the fraction of attached motors, β, which results constant independent of [OM]. On the other hand, in the presence of 1 µM OM $s_0$ decreases from 3.20 to 1.57 nm (Table 1), indicating that the reduction of $T_0$ to ~ 1/2 by 1 µM OM (Fig. 1c) is fully accounted for by a corresponding decrease in $s_0$. $T_0$ and the strain of all the elastic elements decrease in proportion in the whole range of [OM] (Fig. 2e), indicating that

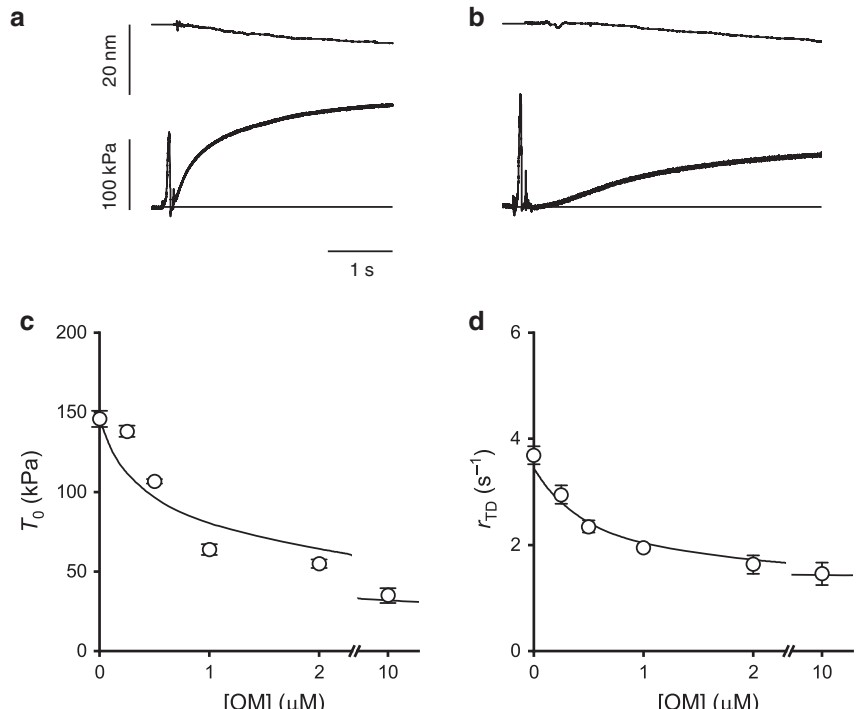

**Fig. 1 Effect of OM on isometric force development.** Force development following T-jump (from 1 to 12 °C) in control (**a**) and in the presence of 1 μM OM (**b**) on an activated fibre (pCa 4.5). Traces show the length change per half-sarcomere (upper trace) and the force (lower trace). The artefact in the traces marks the transition to the test temperature drop. The horizontal line in the lower panels indicates zero force. Fibre length, 4.5 mm; length of the segment under the striation follower, 0.8 mm; average sarcomere length within the segment, 2.3 μm; CSA, 3000 μm2; pCa, 4.5. Dependence of the isometric force $T_0$ (**c**) and of the rate of force development $r_{TD}$ (**d**) on the concentration of OM. Lines, simulation with model of Fig. 6. Data are mean values ± SEM from eight fibres. Source data are provided as a Source Data file.

the reduction of $s_0$ accounts for that in $T_0$. The average force per attached motor ($F_0$) can be calculated from $s_0$ if the stiffness of the myosin motor ($\varepsilon$) is known: $F_0 = s_0 \cdot \varepsilon$. Taking $\varepsilon$ for the slow myosin isoform $= 0.56 \pm 0.04$ pN nm$^{-1}$[19], $F_0$ is $1.8 \pm 0.2$ pN in control and $0.9 \pm 0.1$ pN in 1 μM OM (Fig. 3c).

**OM increases myofilament Ca-sensitivity for motor attachment.** As [Ca$^{2+}$] decreases, the depressant effect of OM on $T_0$ also decreases. As shown in Fig. 3a, at pCa between 4.5 and 6.5, $T_0$ is lower in OM (open circles) than in control (filled circles), while at pCa > 7 $T_0$ is higher in OM. Fitting the data with the Hill equation shows that pCa$_{50}$ (the pCa at which force attains half-maximum) increases from 6.57 in the control (continuous line) to 7.02 in the presence of OM (dashed line) (Table 2), indicating a corresponding increase in Ca$^{2+}$-sensitivity of the force. At the same time the slope of the relation, estimated by the parameter $n$ of the Hill equation, which is an indicator of the degree of cooperativity in myofilament activation, decreases from 2.01 in the control to 0.84 in the presence of OM.

The stiffness of the array of attached motors $e_0$ (an estimate of β) and the average motor strain $s_0$ at each pCa can be calculated both in control and in 1 μM OM using Eqs. (1) and (2) as detailed in "Methods". As shown in Fig. 3b, while at saturating [Ca$^{2+}$] $e_0$ is the same with and without OM, at pCa larger than ~6.5 $e_0$ is higher in OM (open circles) than in control (filled circles). Instead the strain per motor $s_0$ and thus the force per motor $F_0$ are halved by the addition of 1 μM OM at saturating Ca$^{2+}$ (pCa 4.5) (Fig. 3c), but are not affected by the reduction of [Ca$^{2+}$] in either the absence (filled circles) or the presence of OM (open circles).

The larger value of β in OM at sub-saturating [Ca$^{2+}$] but not at saturating [Ca$^{2+}$] is the sign of a higher Ca$^{2+}$ sensitivity, which

underpins OM as a potentiator of contractility[8,29,30]. A more precise definition of the Ca$^{2+}$ dependence of this effect is given in Fig. 3d, where the difference ($\Delta T_0$) between the fitted $T_0$-pCa relations in 1 μM OM (dashed line in Fig. 3a) and in control (continuous line) is plotted versus pCa. At pCa 6.8, the reduced average force per motor (~½) in 1 μM OM is compensated by the increased fraction of attached motors (~twice); at pCa >6.8, 1 μM OM acts as a potentiator increasing the sarcomere force, because, while the force per motor is halved by 1 μM OM, the increased Ca-sensitivity gives a β larger than twice the control.

**Effect of Pi on isometric contraction with and without OM.** The allosteric competition between binding of OM and the presence of Pi in the catalytic site[2,13,14] of the motor has been studied by determining how, at pCa 4.5, the increase of [Pi] (range 1–30 mM, where 1 mM is the [Pi] assumed for the experiments without added Pi[31]) influences the effects of addition of OM on the relevant mechanical parameters of the contraction, $T_0$, $r_{TD}$, $e_0$ (and thus β) and $s_0$ (and thus $F_0$). In control the increase in [Pi] decreases $T_0$ in a dose-dependent manner (Fig. 4a, filled circles) from $156 \pm 1$ kPa in control to $83 \pm 1$ kPa (−47%) in 30 mM Pi, with a [Pi] for half-maximum effect (EPi$_{50}$) of 8 mM. This effect is comparable with that reported in previous papers[23,32,33]. Surprisingly, in 1 μM OM (Fig. 4a, open circles), $T_0$ ($77 \pm 4$ kPa in no added Pi, in these experiments) remains almost constant for Pi < 10 mM and attains a value of $55 \pm 1$ kPa (−30%) in 30 mM Pi. $r_{TD}$ in control (Fig. 4b, filled circles) is $3.8 \pm 0.2$ s$^{-1}$ without added Pi (in agreement with the ordinate intercept in Fig. 1d) and increases with the increase in [Pi] with a slope that progressively decreases. A similar Pi dependence of $r_{TD}$ is observed in the relation determined in 1 μM OM (Fig. 4b, open circles), beyond the depressant effect reported in Fig. 1d.

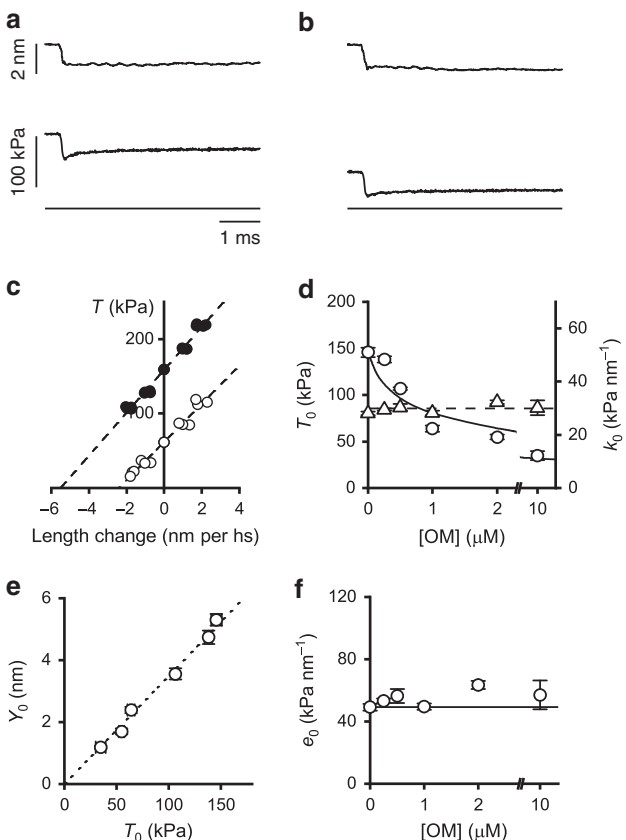

**Table 1 Estimates of the mechanical parameters of the half-sarcomere.**

|  | Control | 1 μM OM |
|---|---|---|
| $T_{0,4.5}$ (kPa) | 156 ± 7 | 69 ± 5 |
| $C_f$ (nm MPa$^{-1}$) | 15.2 ± 1.3 | 15.3 ± 2.0 |
| $s_O$ (nm) | 3.20 ± 0.15 | 1.57 ± 0.10 |

$C_f$ and $s_O$ are determined according to the procedure detailed in "Methods", in control and in the presence of OM. The isometric force at saturating Ca$^{2+}$ is reported in the first row. $T_{0,4.5}$, mean ± SEM from four fibres. $C_f$ and $s_O$, slope and ordinate intercept from data in Fig. 7c. Source data are provided as a Source Data file.

**Table 2 Parameters of Hill's equation fit to force-pCa relation in control and in the presence of OM.**

|  | Control | 1 μM OM |
|---|---|---|
| $n$ | 2.01 ± 0.14 | 0.84 ± 0.15 |
| $pCa_{50}$ | 6.57 ± 0.01 | 7.02 ± 0.07 |

$n$ estimates the slope of the relation, $pCa_{50}$ (the pCa at which force attains half-maximum) estimates the Ca$^{2+}$ sensitivity. Data are mean ± SEM from four fibres. Data from Fig. 3a.

In the presence of 1 μM OM, the lack of the inhibitory effect of the increase in Pi on $T_0$ for [Pi] < 10 mM (Fig. 4a, open circles), while β decreases in the same way as in control (−38% at 10 mM Pi, Fig. 4c), results in the increase of the average motor strain $s_0$ from 1.41 ± 0.15 nm to 2.06 ± 0.06 nm (+46%, left ordinate in Fig. 4d, open circles). Above 10 mM, Pi induces, also in the presence of 1 μM OM, comparable reduction of both $T_0$ and $e_0$ (open circles in Fig. 4a and c), so that $s_0$ remains roughly constant (open circles in Fig. 4d). $F_0$ increases from 0.79 ± 0.10 to 1.16 ± 0.09 pN with the increase in [Pi] in the range 1–10 mM (right ordinate in Fig. 4d, open circles) and then, for [Pi] >10 mM, remains roughly constant at ~1.1 pN, a value that anyway is still lower than in the absence of OM (~1.7 pN).

**OM and the rate of ATP hydrolysis in isometric contraction.** Faced with the contradictory or not exhaustive results on the OM effects on the ATPase activity[2,15,29,35], the effect of the increase in [Pi] (range 1–20 mM) on the ATP hydrolysis rate ($k_{cat}$) in isometric contraction has been determined in the absence and in the presence of OM (Fig. 5a, b). In these experiments, in agreement with the mechanical experiments, in solution with no added Pi $T_0$ is reduced to ~50% the control value by the addition of 1 μM OM (circles in Fig. 5b). $k_{cat}$, instead, does not change significantly (triangles), being 0.055 ± 0.007 mM s$^{-1}$ in the absence of OM and 0.055 ± 0.005 mM s$^{-1}$ and 0.052 ± 0.006 mM s$^{-1}$ in 1 μM and 2 μM OM, respectively.

**Effect of Pi on ATP hydrolysis rate with and without OM.** The increase of [Pi] in the absence of OM reduces by the same amount $T_0$ (Fig. 5c, filled circles, data similar to those in Fig. 4a) and $k_{cat}$ (Fig. 5d, filled circles). This finding is in agreement with previous work[36]. In the presence of 1 μM OM, the increase in [Pi] has an inhibitory effect on $T_0$ that emerges only at [Pi] above 10 mM (Fig. 5c, open circles, data similar to those in Fig. 4a, open circles). Instead, $k_{cat}$ is reduced by the increase in [Pi] (Fig. 5d, open circles) in the same way as in the absence of OM (filled circles). The tension cost of the isometric contraction ($E_T$) can be calculated as the ratio of $k_{cat}$ over $T_0$ (Fig. 5e). Without added Pi, $E_T$ in the presence of 1 μM OM is almost twice the control (points at 1 mM Pi in Fig. 5e, filled circle control, open circle 1 μM OM), as a consequence of the depressant effect of OM on the force per

**Fig. 2 Effect of OM on the relevant mechanical parameters determined by stiffness measurements.** Sample records of the force response (middle trace) to a step length change (upper trace) imposed at the isometric plateau of an activated fibre (pCa 4.5) in control (**a**) and in the presence of 1 μM OM (**b**). Lower traces indicate zero force. **c** $T_1$ relations in control (filled circles) and in the presence of 1 μM OM (open circles) from the same fibre as (**a**) and (**b**). Dashed lines are 1st order regression equations fitted to the data. Same fibre as in Fig. 1. **d** Dependence of $T_0$ (left ordinate, circles) and half-sarcomere stiffness (right ordinate, triangles) on the concentration of OM. Circles are the same data as in Fig. 1c. Continuous line, model simulation of $T_0$–[OM] relation; dashed line, mean value of $k_O$ in the whole range of OM concentrations. **e** Dependence of half-sarcomere strain ($Y_O$) on OM-modulated isometric force ($T_O$). Dotted line, linear regression on data points. **f** Dependence of cross-bridge stiffness ($e_O$) on the concentration of OM. Continuous line, value in the absence of OM. Data points in **d**–**f** are mean values ± SEM from eight fibres. Source data are provided as a Source Data file.

In these experiments, $e_0$ without added Pi ($e_{0,1}$), calculated from the half-sarcomere stiffness $k_0$ (see Supplementary Note 1), is 54.2 ± 4.4 kPa nm$^{-1}$ and 57.5 ± 7.4 kPa nm$^{-1}$ without and with OM, respectively, in agreement with Fig. 2f. $e_0$ decreases with the increase in [Pi] with a slope that progressively reduces in almost the same way without (Fig. 4c, filled circles) and with OM (open circles), attaining ~50% the value without added Pi at 30 mM Pi. Thus, in the absence of OM, $e_0$ (and thus β) reduces with [Pi] almost in proportion to the reduction of $T_0$ (filled circles in Fig. 4a), indicating, in accordance with previous work in skinned fibres from fast skeletal muscle[34], that Pi reduces $T_0$ by reducing β without change in motor strain $s_0$. In these Pi experiments $s_0$ (calculated with Eq. (2) from data as in Supplementary Fig. 1) is 3.08 ± 0.19 nm in the absence of OM (filled circles in Fig. 4d, left ordinate). Consequently, $F_0$ (= $s_0 \cdot \varepsilon$) is 1.75 ± 0.13 pN, independent of [Pi] (right ordinate in Fig. 4d, filled circles).

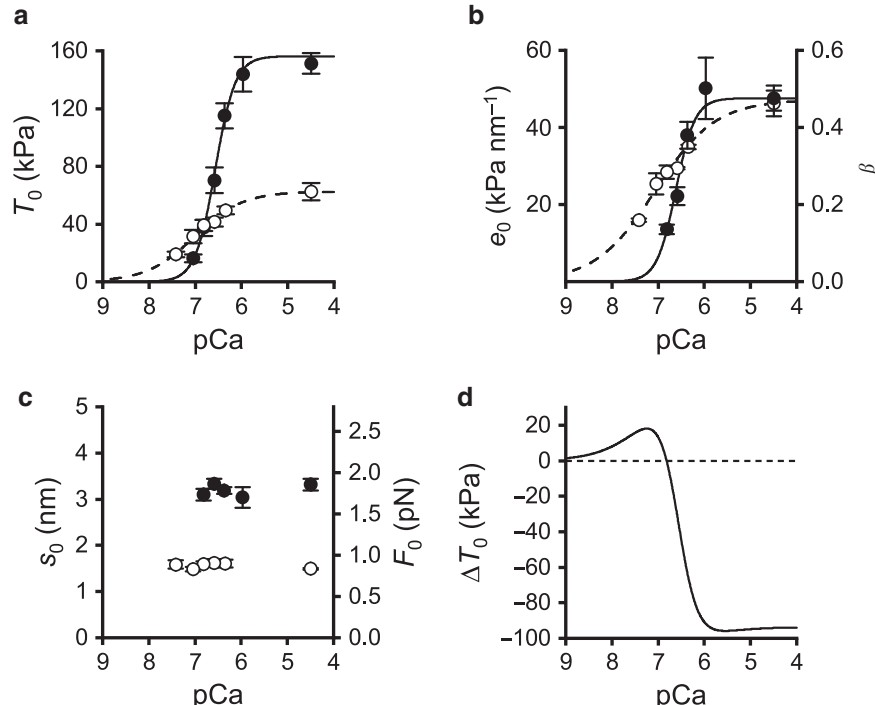

**Fig. 3 Effect of OM on myofilament Ca²⁺ sensitivity for motor attachment. a** $T_0$-pCa relation in control (filled circles) and in the presence of 1 μM OM (open circles). Continuous (control) and dashed (1 μM OM) lines are calculated by fitting Hill equation $T_0 = T_{0,45}/(1 + 10^{n(pCa-pCa_{50})})$ to data. The best fit parameters are listed in Table 2. **b** Dependence of the stiffness of the motor array ($e_O$, left ordinate) and of the fraction of attached motors (β, right ordinate) on [Ca²⁺]. β in control at saturating [Ca²⁺] is assumed 0.47[19]. Continuous (control) and dashed (1 μM OM) lines are calculated by fitting the Hill's sigmoidal equation to data. **c** Dependence of the strain per motor ($s_O$, left ordinate) and of the force per motor ($F_O$, right ordinate) on [Ca²⁺]. **d** Dependence of the difference ($\Delta T_0$) between the fitted $T_0$-pCa relation in 1 μM OM (dashed line in **a**) and in control (continuous line in **a**) on [Ca²⁺]; dashed line indicates zero difference. In all panels, [Ca²⁺] is expressed in pCa units. Data in **a**-**c** are mean values ± SEM from four fibres. Source data are provided as a Source Data file.

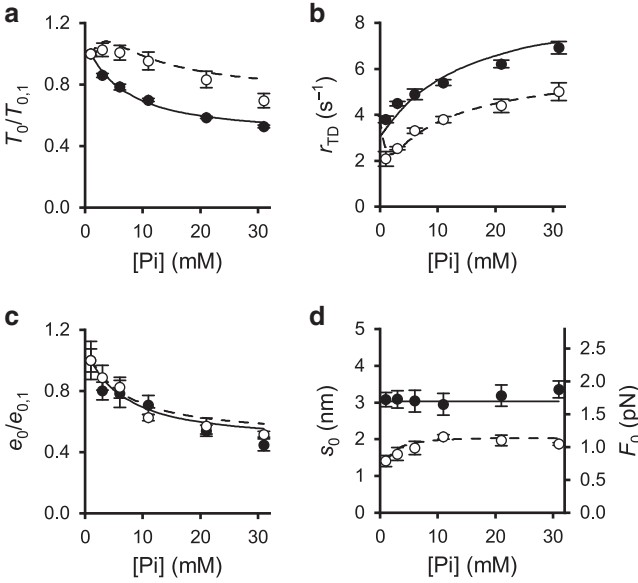

**Fig. 4 Effect of Pi on the isometric mechanical parameters with and without OM.** Filled circles and continuous lines, control; open circles and dashed lines, in the presence of 1 μM OM. **a** Pi dependence of the isometric force ($T_0$) expressed in units relative to the value without added Pi ($T_{0,1}$). **b** Rate of force development ($r_{TD}$). **c** Stiffness of the motor array ($e_O$) relative to the value without added Pi ($e_{O,1}$). **d** Motor strain ($s_O$, left ordinate) and motor force ($F_O$, right ordinate). In each panel: symbols, experimental data; lines, model simulation. Data are mean values ± SEM from three fibres. Source data are provided as a Source Data file.

motor $F_0$ without effect on $k_{cat}$. In the absence of OM $E_T$ remains the same independent of [Pi] (filled circles), while in 1 μM OM (open circles) $E_T$ decreases with the increase of [Pi] approaching the value in the absence of OM.

**Model simulation.** A kinetic model able to explain the effects of OM and Pi on the in situ mechanics and energetics of the β/slow MHC isoform of myosin has been developed. The model is based on that originally proposed for fast skeletal muscle fibres from rabbit psoas[34,37], which has been modified to adapt the rate constants of the forward and backward transitions and the corresponding equilibrium constants ($k_{+x}$, $k_{-x}$ and $K_x$, respectively) to fit the bulk of kinetic data from in vitro and in situ studies of β/slow MHC isoform in control condition (without added Pi and OM)[9,15,32,38] (see Table 3 for the detailed justification of the assumptions made for each kinetic step and the dedicated section in Methods for the criteria of selection of the parameters for the model fitting). In particular, relevant kinetic constraints in this study are as follows: (i) $r_{TD} \sim 4\,s^{-1}$ (Fig. 1d, ten times slower than that of fast muscle), (ii) β at $T_0 = 0.47$ (40% larger than that of fast muscle[19]) and (iii) $k_{cat} = 0.055\,mM\,s^{-1}$ (ten times slower than that of fast muscle[39]). The original model[37] (green-highlighted part of the scheme in Fig. 6), in agreement with experimental evidence[34,39–44], assumes the formation of strongly bound, force-generating motors also before Pi release, so that rise in Pi should exert a minor inhibitory effect on $T_0$ and β than on $k_{cat}$. In contrast to this view, in fast skeletal muscle fibres Pi reduces $T_0$ and β more than $k_{cat}$[34,39], suggesting that the conventional chemo-mechanical cycle has to be integrated with a branch that allows motor detachment at an early stage of the

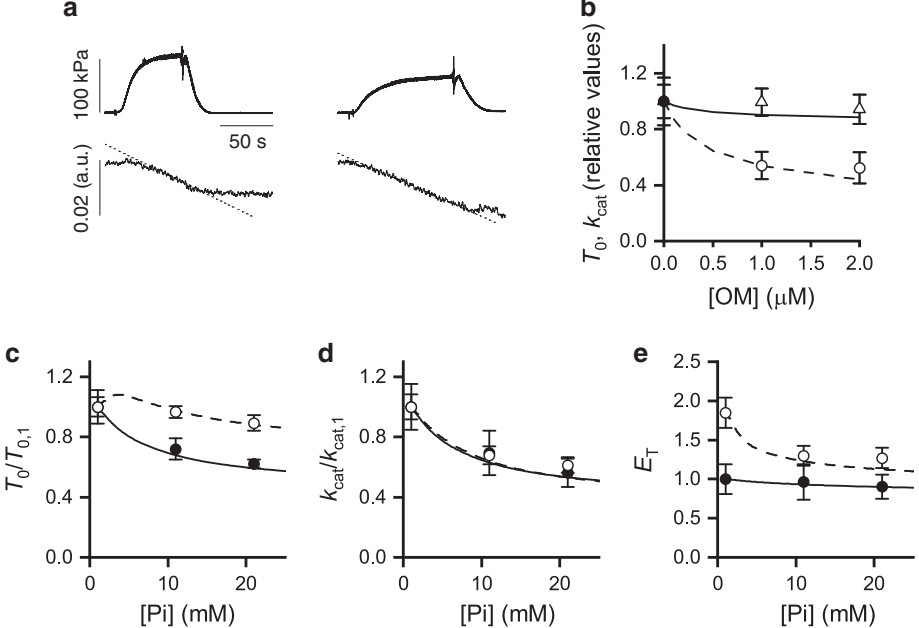

**Fig. 5 Effect of OM on the rate of ATP hydrolysis in isometric contraction and its modulation by Pi. a** Sample records from an activated fibre (pCa 4.5) of the time course of force and light absorbance (in absorbance units, a.u.) in the absence (left) and in the presence (right) of 1 μM OM. The dotted line on the absorbance trace is the 1st order regression equation fitted to the trace after force has attained a steady value. The slope of the regression is used as a measure of the ATP hydrolysis rate ($k_{cat}$). **b** Dependence on OM of $T_0$ (circles and dashed line) and $k_{cat}$ (triangles and continuous line). Values are relative to the isometric values in the absence of OM (98 ± 17 kPa and 0.055 ± 0.007 mM s$^{-1}$, respectively). **c** Dependence on Pi of the isometric force ($T_0$) relative to the value without added Pi ($T_{0,1}$) in the absence (filled circles and continuous line) and in the presence (open circles and dashed line) of 1 μM OM. **d** Dependence on Pi of $k_{cat}$ relative to the value without added Pi ($k_{cat,1}$). Symbols and lines as in (**c**). **e** Tension cost of the isometric contraction ($E_T$). Symbols and lines as in (**c**). $E_T$ is normalized by the value in the absence of OM and no added Pi. **b**–**e** symbols, experimental data; lines, model simulation. Data are mean values ± SEM from 12 fibres in (**b**) and from four fibres in (**c-e**). Source data are provided as a Source Data file.

ATPase cycle (step 6), with Pi still bound to the catalytic site. The finding in this work that β and $k_{cat}$ are reduced by the same extent by the rise in Pi (filled circles in Figs. 4c and 5d) confirms the requirements for the unconventional early detachment with the hydrolysis products still bound to the catalytic site.

To explain the effects of OM on the mechano-kinetics of the β/slow isoform of the myosin motor in situ a branched pathway (pink-highlighted part in the scheme of Fig. 6) is required. In this way the scheme accounts for both the features present in the literature (points *i, ii* below) and those reported in this work (points *iii–v* below): (i) OM binds to the catalytic domain of the myosin motor in the conformation at the beginning of the working stroke (step 8) and increases the probability of its attachment to actin (step 9), accelerating the release of Pi (step 10) and preventing the working stroke and the force generation[2,9,13–15]; (ii) OM-ADP motors detach by a process that does not imply ATP binding (step 12)[9]; (iii) strongly bound force-generating motors and attached OM-bound motors have the same stiffness, to account for the lack of effect of OM on $k_0$ (Fig. 2d, triangles and dashed line) and $e_0$ (Fig. 2f); (iv) the AM′$_{OM}$.ADP state generated after Pi release (step 10) undergoes an isomerization (step 11) to a new state (AM″$_{OM}$.ADP) with increased affinity for Pi and thus at [Pi] > 0 (also without added Pi, when contaminating [Pi] is 1 mM) can rebind Pi, release OM and re-enter the force-generating cycle (step 14); in this way, by appropriate selection of the kinetics of steps 11 and 14 (Table 3), the model is able to account for both the OM-dependent reduction in $r_{TD}$ at any [Pi] > 0 (Fig. 1d, continuous line and Fig. 4b), and the Pi-dependent increase in $F_0$ in 1 μM OM (Fig. 4d, open circles); (v) the rate limiting step in isometric contraction, that in the conventional cycle is the ADP release (step 5), in the OM cycle is detachment (step 12). Thus $k_{+12}$ is set

$\cong k_{+5}$, to account for the finding that $k_{cat}$ is not affected by [OM] (Fig. 5b triangles) and is reduced by the increase in Pi in the same way in the absence as in the presence of OM (Fig. 5d).

In the presence of OM the probability for an M.ADP.Pi motor to escape the conventional cycle (green pathway in Fig. 6) by binding OM and entering the pink pathway (step 8) is controlled by the second-order equilibrium constant $K_8$, which is set at a value $4 \cdot 10^6$ M$^{-1}$ (Table 3) in order to fit the force—[OM] relation of Fig. 1 (continuous line). The kinetics of steps 11, the isomerization that changes Pi affinity of the OM-bound motor, and step 14, in which Pi competes with OM for binding to the attached motor after the isomerization step (step 11), are dictated by the following constraints: (i) without added Pi ([Pi] ~1 mM) and in the presence of OM, $r_{TD}$ attains the minimum value ~½ the control with a [OM]$_{50}$ = 0.5 μM (Fig. 1d); (ii) increase in [Pi] reduces β in the same way in the absence and presence of OM (Fig. 4c); (iii) in 1 μM OM the force per motor increases by 50% with the increase of [Pi] in the range 1–10 mM (Fig. 4d). These results are fitted (dashed line) assuming, for step 11 an equilibrium constant ($K_{11}$) = 1, and for step 14, a second-order rate constant for Pi binding ($k_{+14}$) = $3 \cdot 10^3$ M$^{-1}$ s$^{-1}$ and a second-order rate constant for OM binding ($k_{-14}$) = $3 \cdot 10^6$ M$^{-1}$ s$^{-1}$ (Table 3).

## Discussion

Sarcomere-level mechanics of demembranated fibres from rabbit soleus muscle allow a double mechanism of OM action on the β/slow MHC isoform to be revealed, which eventually accounts for a Pi-dependent recovery of efficiency of the contractile response of OM-treated fibres.

The first mechanism of action, which confirms the conclusions of recent work in vitro[9,14], implies a dose-dependent depression

**Table 3 Rate constants for the forward ($k_{+x}$) and backward ($k_{-x}$) transitions and corresponding equilibrium constants ($K_x$) for the scheme of Fig. 6.**

| Step in the cycle | $k_{+x}$ | $k_{-x}$ | $K_x$ | Justification |
|---|---|---|---|---|
| **A, green-highlighted path in Fig. 6** | | | | |
| (1) | $1 \times 10^6\,M^{-1}s^{-1}$ | $1\,s^{-1}$ | $1 \times 10^6\,M^{-1}$ | According to refs. [32,38,62,63] |
| (2) | $14\,s^{-1}$ | $9\,s^{-1}$ | 1.56 | $k_2$, according to ref. [38]; $k_{-2}$, selected to adjust the distribution of detached myosin motors |
| (3) | $2.2\,s^{-1}$ | $4.5\,s^{-1}$ | 0.49 | $k_{+3}$, adjusted to fit the fraction of attached motors in isometric contraction[19] |
| (4) | $11\,s^{-1}$ | $0.9 \times 10^3\,M^{-1}s^{-1}$ | $12.2 \times 10^{-3}\,M$ | $k_{+4}$ from refs. [9,15] (assuming $Q_{10} = 2$) and $K_4$ adjusted to fit the Pi dependence of the rate of force development assuming these steps are 50 times slower than in psoas muscle[32] |
| (5) | $1\,s^{-1}$ | $1 \times 10^3\,M^{-1}\,s^{-1}$ | $1 \times 10^{-3}\,M$ | Adjusted to fit $k_{cat}$ in isometric contraction in the absence of OM[64-66] |
| (6) | $0.8\,s^{-1}$ | $2 \times 10^{-3}\,s^{-1}$ | $0.4 \times 10^3$ | Adjusted to fit the Pi dependence of $k_{cat}$ in the absence of OM[37] |
| (7) | $50\,s^{-1}$ | – | – | Rapid and irreversible release of hydrolysis products and ATP binding[37] |
| **B, pink-highlighted path in Fig. 6** | | | | |
| (8) | $400 \times 10^6\,M^{-1}s^{-1}$ | $100\,s^{-1}$ | $4 \times 10^6\,M^{-1}$ | Adjusted to have the correct proportion of OM-bound myosin motors |
| (9) | $1.2\,s^{-1}$ | $1\,s^{-1}$ | 1.2 | $k_{+9}$ and $k_{-9}$ adjusted to account for OM-dependent increase of probability of actin attachment by motors ($K_9 > K_3$)[15] and for the constant total number of attached motors in the presence of OM |
| (10) | $18\,s^{-1}$ | $1.48 \times 10^3\,M^{-1}\,s^{-1}$ | $12.2 \times 10^{-3}\,M$ | $k_{+10}$ set to 1.64 $k_{+4}$, to take into account the effect of OM on the rate of Pi release[9,15]; $K_{10}$, same value as in step (4) |
| (11) | $1000\,s^{-1}$ | $1000\,s^{-1}$ | 1 | Rapid isomerization step; $K_{11}$ set to 1 to adjust the stoichiometry of the Pi effect |
| (12) | $1.5\,s^{-1}$ | $2 \times 10^{-3}\,s^{-1}$ | $1 \times 10^3$ | Detachment set to fit $k_{cat}$ in the presence of OM |
| (13) | $3\,s^{-1}$ | – | – | Irreversible release of ADP and OM followed by rapid ATP binding |
| (14) | $3 \times 10^3\,M^{-1}s^{-1}$ | $3 \times 10^6\,M^{-1}s^{-1}$ | $1 \times 10^{-3}$ | Adjusted to fit the kinetics of force development and the Pi dependence of $T_0$ and $e_0$ in the presence of OM |

Values for the model in the absence of OM (A, green-highlighted in Fig. 6) and for the branched pathway induced by OM (B, pink-highlighted in Fig. 6). The justification of the kinetic parameters are reported in the last column. Where possible, the parameters are selected according to the relevant information from the literature. The selection of the value of the kinetic parameters concerning OM and OM-Pi actions and the sensitivity of the model fit to the selected value are achieved with an iterative procedure of model simulation by changing one parameter at a time as detailed in "Methods" and in Supplementary Figs. 2 and 3. $x$ represents the step number, as reported in brackets in the first column. For the second-order rate constants, the apparent rate constants are calculated assuming [MgATP] = 5 mM and [ADP] = 20 μM; [Pi] and [OM] according to the experimental conditions.

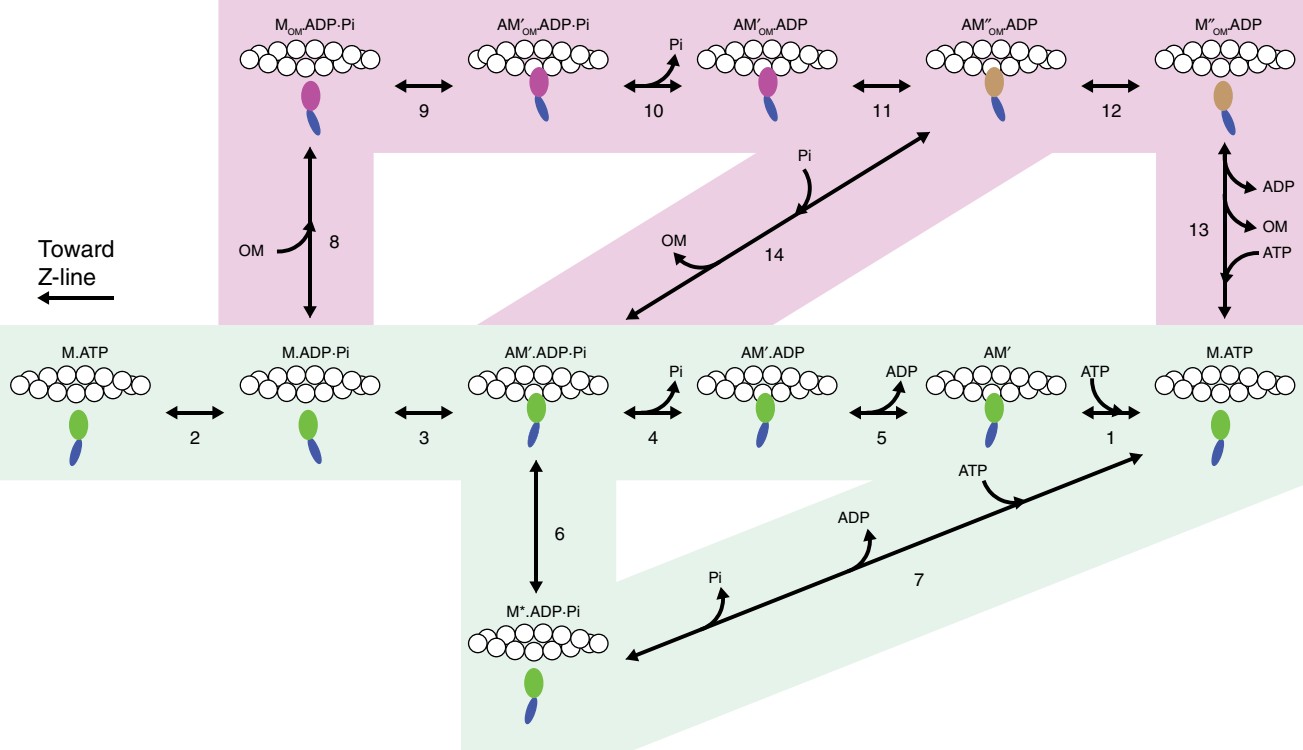

**Fig. 6 Chemo-mechanical cycle of the myosin motor.** The green-highlighted part is the original model[37]; the pink-highlighted part is the branched pathway promoted by OM. Myosin catalytic domain is green for the motors without OM (M) and magenta and brown for the OM-bound motors ($M_{OM}$) before and after the isomerization step (step 11), respectively. The working stroke in the attached head is indicated by the tilting of the light chain domain (the lever arm, blue) towards the Z-line. The system of linear differential equation for the calculation of state transitions is reported in Supplementary Note 2.

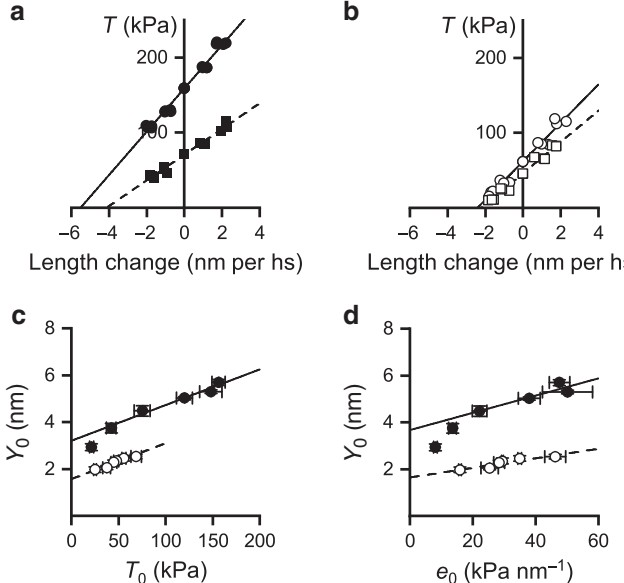

**Fig. 7 Estimate of myofilament compliance and motor strain.** $T_1$ relations from a fibre in control (**a**) and in the presence of 1 µM OM (**b**) at pCa 4.5 (circles) and 6.59 (squares). The relations are obtained by plotting the extreme force attained at the end of the length step, $T_1$, versus the step size. Lines in **a** and **b** are 1st order regression equations fitted to the data (pCa 4.5, continuous; pCa 6.59, dashed). The abscissa intercept of each line is the average strain of the half-sarcomere before the length step ($Y_0$). Fibre length, 4.2 mm; segment length under the striation follower, 0.77 mm, average segment sarcomere length, 2.34 µm, test temperature, 12.2 °C. **c** Relations of $Y_0$ versus $T_0$ at different pCa determined in control (filled circles) and in the presence of 1 µM OM (open circles); mean values ± SEM from four fibres. The lines are linear regressions on the pooled data for forces >30 kPa in control (continuous) and in the presence of OM (dashed). The slope and the ordinate intercept of the linear regressions are listed in Table 1. **d** Relations of half-sarcomere strain ($Y_0$) versus the stiffness of the myosin motors at different pCa ($e_0$) in control (filled circles) and in the presence of 1 µM OM (open circles); mean values ± SEM from four fibres. The lines are linear regressions on the pooled data for stiffness > 20 kPa nm$^{-1}$ in control (continuous) and in the presence of OM (dashed). **c** and **d**, data from four fibres. Source data are provided as a Source Data file.

of the active sarcomere force by OM binding to the myosin motor in the pre-working stroke state that promotes an unconventional cycle in which the working stroke is prevented and the detachment occurs without binding of ATP.

We find that, irrespective of [Ca$^{2+}$], in 1 µM OM the fraction of OM-bound attached motors is ~50% of the total (β), as indicated by the average force per motor, $F_0$, that remains ~50% of that in control at any [Ca$^{2+}$] tested (Fig. 3c). The generation by OM of a pathway leading to strongly actin-bound motors in parallel with that responsible for the conventional force-generating cycle increases β at sub-saturating [Ca$^{2+}$] but not at saturating [Ca$^{2+}$] (Fig. 3b). This increase in Ca$^{2+}$ sensitivity, measured by the increase of pCa$_{50}$ by 0.4 units (Table 2), is similar to that reported for demembranated cardiac myocytes[8,29,30] and is attributable to the contribution of OM-motors to the cooperative effect of attached motors in switching ON the thin filament[9,45]. The potentiating role of this mechanism in relation to the level of Ca$^{2+}$ is quantified by plotting the difference between $T_0$ in 1 µM OM and in control ($\Delta T_0$) versus pCa (Fig. 3d): $\Delta T_0$ is negative at saturating and high [Ca$^{2+}$] and crosses zero at pCa 6.8 because, at this pCa, the reduced average force per motor (~½) in 1 µM OM is compensated by the

increased total number of attached motors (~twice); at any pCa > 6.8, 1 µM OM acts as a potentiator increasing the sarcomere force, because the increased Ca-sensitivity makes the total number of motors larger than twice the control. This mechanism applies equally well to explain the OM potentiating effects at the partial Ca$^{2+}$-activation found in demembranated cardiac myocytes[8,29,30]. The force-pCa relation in 1 µM OM is characterized not only by the larger Ca-sensitivity but also by a reduced slope: Hill coefficient $n$ is reduced by a factor of 2 (from 2.01 to 0.84, Table 2), in agreement with previous work on both demembranated cardiac myocytes and slow-skeletal muscle fibres[8,29,30]. The explanation could be related to a negative effect on the cooperativity index of OM-bound motors if the propagation of the activation along the thin filament induced by motor attachment depends on the motor force.

The second mechanism of action of OM, which is revealed in this work, is the recovery from motor force depression with increase in [Pi] in the range 1–10 mM. This finding per se scales back the relevance of previous mechanisms of action of OM defined in the absence of added Pi, since 1–10 mM is the physiological range of [Pi] in working muscle. The molecular basis of this mechanism is implicit in the effect of OM on $r_{TD}$ also in the absence of added Pi ([Pi] = 1 mM): $r_{TD}$ is depressed by OM with the same dose dependency as $T_0$ (Fig. 1 c, d), with [OM] for half-maximum effect 0.5 µM, which noteworthy is in the range (0.1–0.6 µM) of the plasma [OM] in patients under therapeutic treatment[3,5,46]. This consideration suggests further work in which the mechanism revealed here is tested also for the more clinically competent range of [OM].

The kinetic model of Fig. 6 is able to fit both $T_0$- and $r_{TD}$-OM relations (continuous lines in Fig. 1 c and d, respectively) assuming that, after Pi release, the OM-bound motor in the AM′$_{OM}$.ADP state undergoes an isomerization to an AM″$_{OM}$.ADP state with higher affinity for Pi favouring an "allosteric" exchange Pi-OM (step 14), which allows the motor to re-enter the force-generating cycle. Noteworthy, it is not possible to simulate the depressant effect of OM on $r_{TD}$ confining the allosteric competition between OM and Pi to step 10, because the OM-dependent effect on force would not imply any reduction in $r_{TD}$. Only introducing the pink-highlighted path that contributes to $T_0$ with a slower process through the isomerization (step 11) and the following OM-Pi exchange (step 14) the time course of $T_0$ is slowed and the $r_{TD}$-Pi relation in 1 µM OM is shifted downward (Fig. 4b). The stoichiometry of the Pi-dependent recovery of the average force per motor in the presence of 1 µM OM (Fig. 4d, open circles) is mainly accounted for by the equilibrium constants of steps 11 (1) and 14 (10$^{-3}$). For step 14, assuming a second-order rate constant for Pi binding three orders of magnitude smaller than the second-order rate constant for OM binding, the equilibrium is near the unity when the ratio between [Pi] and [OM] is about 10$^3$ (Table 3), that is, with the near millimolar physiological concentration of Pi, at micromolar [OM].

The energetics of the combined OM-Pi action on contractility has been defined by determining the effect of OM on the rate of ATP hydrolysis ($k_{cat}$) during isometric contraction at saturating Ca$^{2+}$ and its modulation by Pi. Increase in [OM] up to 2 µM, which decreases $T_0$ to ½ the control, does not change $k_{cat}$ (Fig. 5b). This result is specific for a loaded contraction, as in both solution[15,35] and low-load laser trap experiments[9] on β-cardiac myosin the ATPase rate has been found to be reduced in OM with respect to the control.

In either the pink-highlighted or the conventional green-highlighted path under high load the motors accumulate in the ADP-bound state because either OM (pink path) or the high load (green path) limit the probability to go through the working stroke that would accelerate the next step, the conformation

dependent ADP release[44,47–49]. Consequently, independent of the reason that limits the execution of the working stroke, both step 12 for OM-bound motors and step 5 for OM-free motors are slow (Table 3) and the ATPase rate is limited in the same way independent of the proportion of motors passing through either path. In this way the model can simulate the OM-dependent drop in $T_0$ (dashed line in Fig. 5b) without effect on $k_{cat}$ (continuous line).

In the absence of added Pi, OM reduces $T_0$ through a reduction in $F_0$ at constant $k_{cat}$, thus it increases the tension cost ($E_T$) of the isometric contraction. One micromolar OM, which halves $F_0$, doubles $E_T$ (Fig. 5e). Noteworthy, this conclusion extends from maximal contractions at saturating [$Ca^{2+}$] to contractions at sub-saturating [$Ca^{2+}$] because reduction in [$Ca^{2+}$] reduces the total number of attached motors (Fig. 3b) without affecting $F_0$ (Fig. 3c).

A direct test of Pi effect on OM-bound motors at pCa ∼7, where OM exerts its potentiating action via the increase in $Ca^{2+}$ sensitivity (Fig. 3d), would be quite important, considering that the internal [$Ca^{2+}$] in the heart systole ranges 0.2–0.6 μM[50,51]. However, the sarcomere-level mechanics on isometric contraction of skinned soleus fibres activated at pCa ∼7 would fail to produce data that can be interpreted in terms of Eq. (1), because the stiffness of the motor array would be too low (<20 kPa nm$^{-1}$, Fig. 3b) and the half-sarcomere compliance analysis would be complicated by the significant contribution of the additional elasticity in parallel with the motor array (see "Methods", Fig. 7b). In this respect it must be noted also that the nature and the effects of the additional parallel elasticity are likely different in the slow-skeletal muscle and in the cardiac muscle, which have different isoforms of the accessory and cytoskeleton proteins like My-BPC and titin. Consequently, the conclusions from soleus muscle could not be directly translated to cardiac muscle as it is instead possible under the condition selected in this study (stiffness of the motor array > 20 kPa nm$^{-1}$), in which the half-sarcomere compliance is only determined by the number of β/slow myosin isoforms attached to the actin filament.

Under the conditions defined above, we provide the evidence for a previously unknown inotropic effect on the β/slow MHC isoform in OM-treated myocytes: increase in [Pi] within the physiological range (1–10 mM) recovers from the inhibitory effect of OM on $T_0$ by increasing the proportion of motors re-entering the force-generating cycle and this, at constant $k_{cat}$, implies a corresponding reduction in tension cost (Fig. 5e). This effect of Pi on the OM-bound motors could represent an energetically efficient response in OM-treated cardiomyopathic patients, whenever the increase of heart-beat frequency induces an increase in [Pi][27,28].

## Methods

**Animal and ethical approval.** Experiments have been done on demembranated fibres from the soleus muscle of adult male New Zealand white rabbits at either the PhysioLab Research Unity of the Biology Department of the University of Florence (mechanical experiments), or the Department of Physiology of the VU University Medical Center in Amsterdam (ATPase rate measurements). The experiments were carried out according to the protocols approved by the Ethical Committee of the University of Florence and by the Italian Ministry of Health (authorization n. 956/2015 PR) in compliance with the Italian regulation on animal experimentation, Decreto Legislativo 26/2014 and the EU regulation (directive 2010/63). Rabbits (4–5 kg weight, 20–30 weeks old) were sacrificed by injection of an overdose of sodium pentobarbitone (150 mg kg$^{-1}$) in the marginal ear vein. Two rabbits were used for this work. All animals have been kept with free access to food and water prior to use.

**Fibre preparation and mechanical apparatus.** Small bundles (40–80 fibres) of soleus muscle fibres were stored in skinning solution containing 50% glycerol at −20 °C for 3–4 weeks and single fibres were prepared just before the experiment as already described[18,52]. The osmotic agent Dextran T-500 was added to all solutions at the concentration of 4 g/100 ml (4% weight/volume) that is known to reverse the permeabilization-induced increase of interfilamentary spacing and cross-sectional area (CSA) to the value before skinning[18,53–57].

A fibre segment, 4–6 mm long, was mounted between the lever arms of a loudspeaker motor able to impose steps in length complete within 80 μs and a capacitance force transducer with resonant frequency 40–50 kHz[18]. Sarcomere length ($sl$), width ($w$) and height ($h$) of the fibre were measured at 0.5 mm intervals in the 3–4 mm central segment of the relaxed fibre with a 40× dry objective (Zeiss, NA 0.60) and a 25× eyepiece. The fibre length ($L_0$) was adjusted to have a $sl$ within the range 2.3–2.5 μm. The fibre CSA was determined assuming the fibre cross-section as elliptical (CSA = π/4·$w$·$h$) and its value ranged between 2100 and 5300 μm². Fibres were activated by temperature jump using a solution exchange system[18]. The fibre was kept in the activating solution at the test temperature (12 °C) for 4–5 s for the mechanical measurements. A striation follower[58] allowed nanometre-microsecond resolution recording of length changes in a selected population of sarcomeres (range 500–1200 sarcomeres) starting at the time the optic path was permitted through the glass window in the floor of the test temperature drop (see ref. [18] for details).

**Half-sarcomere stiffness measurements.** Step length changes (ranging from −2 to +2 nm per hs, stretch positive, rise time 100 μs) were imposed on the isometrically contracting fibre to estimate the half-sarcomere stiffness by the slope of the relation between the force attained at the end of the step and the change in half-sarcomere length in the sarcomere population of the segment monitored by the striation follower ($T_1$ relation) (Figs. 2a, b and 7). To enhance the precision of stiffness measurements, a train of different-sized steps at 200-ms intervals was applied during each activation and, to maintain the isometric force before the test step constant, each test step was followed, after a 50-ms pause, by a step of the same size but opposite direction[18,19]. Half-sarcomere stiffness measurements were done (i) at different [OM] (range 0.1–10 μM) in the absence of added Pi (that is with [Pi] ∼1 mM) and with [$Ca^{2+}$] for maximum activation (pCa 4.5), (ii) at different [$Ca^{2+}$] (range of pCa 4.5–7.5) without and with 1 μM OM and (iii) at different [Pi] (range 1–20 mM) at pCa 4.5 without and with 1 μM OM.

**Half-sarcomere compliance analysis.** The elastic characteristics of the array of myosin motors during active isometric contraction can be estimated from the stiffness of the half-sarcomere with a mechanical protocol that allows to isolate and subtract the contribution of the filament compliance ($C_f$) to half-sarcomere compliance ($C_{hs}$, which is the reciprocal of the half-sarcomere stiffness $k_0$). The protocol consists in determining the $T_1$ relations during activations at different [$Ca^{2+}$]. In fact, as previously demonstrated[18], [$Ca^{2+}$]-dependent changes in maximum isometric force ($T_0$) are fully accounted for by changes in the number of attached motors. Under this condition the relation between the half-sarcomere strain ($Y_0 = C_{hs} \cdot T_0$) and $T_0$ can be interpreted with a simple mechanical model of the half-sarcomere in which the attached motors are represented by an array of in parallel springs, the number of which, but not the strain ($s_0$), changes in proportion to force, in series with the actin and myosin filaments represented by a spring, which accounts for an equivalent filament compliance ($C_f$) (Model 1 in ref. [59]). According to Model 1, $C_{hs}$, is expressed by the equation[59]:

$$C_{hs}(=1/k_0) = C_f + 1/e_0, \qquad (1)$$

where $1/e_0$, the compliance of the array of motors, is the reciprocal of the stiffness of the array ($e_0$, which is proportional to the number of attached motors). From the above equation the strain of the half-sarcomere $Y_0$ ($= C_{hs} \cdot T_0$) can be derived:

$$Y_0 = (C_{hs} \cdot T_0 = C_f \cdot T_0 + T_0/e_0 =)C_f \cdot T_0 + s_0, \qquad (2)$$

where $s_0$ is the average strain in the attached motors at $T_0$. According to the model, $Y_0$ increases linearly with the force with a slope that is explained by the increase in the strain of the myofilaments with constant compliance $C_f$, while $s_0$, estimated by the ordinate intercept of the $Y_0 - T_0$ relation, is the same independent of $T_0$.

The $T_1$ relations were determined at different pCa in control and in the presence of 1 μM OM (which reduces $T_{0,4.5}$ to ½). Figure 7 shows the relations obtained at pCa 4.5 (circles) and 6.59 (squares) in control (a) and in the presence of 1 μM OM (b). The half-sarcomere stiffness ($k_0$, estimated by the slope of the linear fit to data, dashed line) reduces, lowering [$Ca^{2+}$], from 28.8 ± 1.0 kPa nm$^{-1}$ (filled circles) to 17.1 ± 0.7 kPa nm$^{-1}$ (filled squares) in control (a) and from 26.3 ± 1.5 kPa nm$^{-1}$ (open circles) to 20.9 ± 0.9 kPa nm$^{-1}$ (open squares) in the presence of OM (b). In both conditions the half-sarcomere strain at $T_0$ ($Y_0$, estimated by the extrapolation of the fit to zero force) is shifted rightward with lowering [$Ca^{2+}$], but less than in proportion to the reduction of the ordinate intercept, in agreement with the reduction of $k_0$. In Fig. 7c, $Y_0$ at different [$Ca^{2+}$] in control (filled circles) and in the presence of 1 μM OM (open circles) is plotted against the corresponding $T_0$. In control, $Y_0$ increases linearly with the isometric force, with the exception of the lowest points (for $T_0 < 50$ kPa) that are shifted downward. The linear relation is expected from Model 1 and the deviation from linearity at very low forces has been explained with the presence of an elastic element in parallel with the myosin motors with a stiffness that is so small that it affects the linear relation only in the region where the number of the motors, and thus their cumulative stiffness are quite low[59]. Noteworthy, the $Y_0 - T_0$ relation in OM (open circles in Fig. 7c) does not show deviation from linearity (see below for the explanation). The values of $C_f$ and $s_0$, estimated by fitting Equation (2) to $Y_0 - T_0$ values in control (in the range

of $T_0 > 50$ kPa) and in the presence of OM are reported in Table 1. The value of $C_f$ in the presence of OM (15.3 nm MPa$^{-1}$) does not differ significantly from the control value (15.2 nm MPa$^{-1}$, $P > 0.9$, Student's $t$ test), indicating that also in this case the dependence of $T_0$ on [Ca$^{2+}$] is explained by the Ca$^{2+}$-modulation of the number of attached motors without change in motor strain.

In OM the $Y_0 - T_0$ relation (open circles in Fig. 7c) does not show deviation from linearity for the same low forces at which the control relation shows a downward shift. This apparent contradiction is explained by considering the effect of OM that reduces the force per motor but increases the Ca-sensitivity for motor attachment, so that at the same low values of $T_0$, in the presence of OM, the number of attached motors and thus their cumulative stiffness are higher than in control and still in the range of values that mask the contribution of the parallel elastic element. A quantitative evaluation of this interpretation is obtained in Fig. 7d by plotting $Y_0$ versus the stiffness of the array of motors $e_0$ at each pCa (data from Fig. 3b) for both control (filled circles) and OM (open circles). It can be seen that both control and OM relations are almost linear in the same range of $e_0$, where the stiffness of attached motors is high enough to mask the contribution of the parallel elastic element, and that the control relation deviates from linearity, uncovering the presence of a parallel elastic element, for the small $e_0$ values that are not attained by the OM relation.

**ATPase measurements**. Fibre bundles prepared in Florence lab as described above were transported to Amsterdam in ice-cold relaxing solution. ATPase activity (temperature 12 °C) at saturating [Ca$^{2+}$] and at different [Pi] was measured photometrically by enzymatic coupling of the regeneration of ATP to the oxidation of NADH present in the bathing solution[36,60,61] both in control and in the presence of OM (1 and 2 μM). NADH breakdown was monitored via the absorption of near UV light at 340 nm. Briefly, the set up consisted of two anodized aluminium troughs (volume 80 μl each) containing relaxing and pre-activating solution and a measuring chamber (volume about 30 μl) containing activating solution. The solution was continuously stirred via a membrane at the base of the chamber. UV light passed through the chamber beneath the fibre, and the transmitted light was monitored by two UV-enhanced photodiodes at 340 and 400 nm. The photodiode at 400 nm provided a reference signal, independent of NADH concentration. A syringe, controlled by a stepper motor, was used to add 0.05 μl of 10 mM ADP at the end of each recording to calibrate the absorbance signal. Force during the enzymatic assay was measured by means of a strain gauge transducer (AE801 SensoNor, Horten, Norway). The force and the absorbance signals were filtered at 1 kHz and 2.5 Hz, respectively. The ATPase rate was measured as the slope over time of the absorbance signal.

**Procedure for the selection of the parameters of the kinetic model**. The procedure followed to assign a specific value to the kinetic parameters concerning the effects of OM and Pi on the force ($T_0$), rate of force development ($r_{TD}$), stiffness of the motor array ($e_0$) and ATPase rate ($k_{cat}$) (eight relations altogether) is based on an iterative routine in which the values of the rate constants (or equilibrium constants) controlling the kinetics of the relevant steps (8–12 and 14) are changed one at a time. For each value of the rate/equilibrium constant ($k$) the weighted mean of residual errors for each relation is calculated according to the expression:

$$R_w(k) = \frac{1}{N} \sum_{i=1}^{N} \frac{(x_i - xmod(k)_i)^2}{x_i^2},$$

where $R_w(k)$ is the mean sum of residuals (weighted for the squared value of the experimental point to allow homogeneous comparison among the eight different relations), $x_i$ is the experimental point value, $xmod(k)_i$ is the corresponding simulated value for a given $k$, and $N$ is the number of points in each relation. An example of the selection procedure for a given rate constant ($k_{+14}$) in relation to its effect on each of the eight relevant relations is given in Supplementary Fig. 2, where in the left column six simulations obtained for different values of $k_{+14}$ (each identified by a different colour) of four of the eight relations are superimposed on the experimental relations (open symbols) and, in the right column, the weighted mean of residuals ($R_w$) are plotted versus the value of $k_{+14}$ assumed in each simulation (identified by the same colour as in the left column). As shown by the plots in the right column of Supplementary Fig. 2, the sensitivity of the test in showing a minimum mean residual and its dependence on the value of the rate/equilibrium constant varies depending on the relation. For this, the criterium of selection of the best value for any given rate/equilibrium constant is based on the minimization of the weighted mean of the residuals averaged over the eight relations (global mean, $\langle R_w(k) \rangle$):

$$\langle R_w(k) \rangle = \frac{1}{8} \sum_{p=1}^{8} R_w(k).$$

In Supplementary Fig. 3 the values of $\langle R_w(k) \rangle$ are plotted versus the rate/equilibrium constant under consideration. The value selected with this global fitting for each rate/equilibrium constant, identified by the abscissa of the filled symbol, is reported in Table 3.

**Solutions**. The composition of the solutions, similar for the mechanical and the ATPase measurements (MgATP, 5 mM; free Mg$^{2+}$, 1.2 mM) is reported in Supplementary Table 2. The activating solution at a given pCa (range 7.5–4.5) was obtained by mixing relaxing and activating solutions. OM (CK-1827452 from Selleckchem, USA) was dissolved in DMSO (Sigma D-5879) (stock solution, 17.5 mM). Different OM concentrations in the final solutions (0.1–10 μM range) were obtained by partial dilution starting from the stock solution. Four percent dextran was added to all solutions for the mechanical experiments.

**Statistical analysis**. All data were analyzed using dedicated programs written in LabVIEW (National Instruments) and Microsoft Excel and Origin 2018 (OriginLab Corp., Northampton, MA, USA) software. Error bars on mean data points are ±SEM.

For mean data in the figures and tables, the number of fibres is reported in the legends and the number of observations, which may vary for each data point, can be recovered from the Source Data files.

**Reporting summary**. Further information on research design is available in the Nature Research Reporting Summary linked to this article.

## Data availability
The authors declare that the data supporting the findings of this study are available within the paper and its Supplementary Information files. The source data for Figs. 1, 2, 3, 4, 5, 7, Supplementary Figs. 1, 2 and 3 and for Tables 1, 2 and Supplementary Table 1 are provided as a Source Data file. All remaining data will be available from the corresponding author upon reasonable request. Source data are provided with this paper.

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

## Acknowledgements

We thank Massimo Reconditi for insightful comments on the manuscript and valuable support to the definition of the procedure for the optimization of the simulation and the staff of the mechanical workshop of the Department of Physics and Astronomy (University of Florence) for mechanical engineering support. This work was supported by the University of Florence and Fondazione Cassa di Risparmio di Firenze (Italy).

## Author contributions

V.L., M.L., G.P., M.C., C.O. and G.S. designed the research. S.G., M.C. and I.P. performed the experiments and analysed the data. V.L., M.L. and G.P. wrote the paper or revised it critically for important intellectual content. All authors participated in discussions on this work and approved the final version of the manuscript.

## Competing interests

The authors declare no competing interests.
