## [Peer Review File · Nature Communications]

Reviewers' comments:

Reviewer #1 (Remarks to the Author):

Summary: The authors of the submitted manuscript present data investigating the mechanism by which the systolic heart failure (HF) drug, Omecamtiv Mecarbil (OM), affects the force output and cycling kinetic of rabbit soleus muscle in situ. Mechanical experiments confirmed previously published results that OM increases force at submaximal $[Ca^{2+}]$, decreases force near maximal $[Ca^{2+}]$, increases calcium sensitivity, and decreases cooperativity. Using a simplified spring-model of the sarcomere, the authors show that the OM-induced reduction in force at max $[Ca^{2+}]$ can be attributed to the reduction in the average strain experienced by each attached motor, as opposed to a reduction in the number of bound motors. At submaximal $[Ca^{2+}]$, OM-induced force increases were attributed to an increase in the number of bound motors, which supports previous findings. Interestingly, the addition of physiological concentrations of inorganic phosphate (Pi) significantly mitigates the OM-induced force reduction at maximal $[Ca^{2+}]$. The addition of Pi also had significant dose dependent effects on the rate of force redevelopment and the average motor strain. As a second source of kinetic insight, the authors measured the overall ATPase rate (kcat) in isometrically contracting muscle to find that OM did not significantly impact the Pi dependence of kcat but did reduce the muscle's tension cost. As the authors mention, this finding is specific to the context of high load contraction. To explain the observed effects of OM and Pi, the authors extend a previously published kinetic model of muscle myosin. The main feature of the model is the addition of an OM pathway that inhibits the power stroke and introduces an isomerization step following $P \rightarrow i$ where the new state has an increased Pi affinity. The new state allows for competitive interactions between OM and $P-i$ in which $P-i$ can allosterically displace OM, thus allowing the motor to reenter the usual force producing ATPase cycle. Finally, the authors suggest that their experimental findings and model predictions may identify an energy-efficient response to OM during exercise when intercellular Pi increases within the heart.

Major Comments:

1. While it is understandable from the experimental perspective why OM's effects were studied in soleus skeletal muscle instead of cardiac muscle, the authors translate results between the two systems without much discussion of potential limitations and caveats. Such translation may be appropriate for single molecule results as the dominant myosin isoform in soleus and cardiac muscle are so similar (ref 1) but this is harder to justify at the fiber level where other important sarcomeric proteins differ in isoform and functional role. Furthermore, considering that the vast majority of fiber level studies on OM are in cardiac muscle and that little has been previously published on OM's effect on soleus skeletal muscle, a detailed limitations section and/or additional experimental evidence is needed to contextualize/justify the translation from soleus to cardiac muscle.

2. In the introduction (manuscript lines 63-69) the authors use the fact that intracellular $[Ca^{2+}]$ remains below saturating levels during left ventricular systole in their proposed mechanism explaining how OM and Pi jointly impact muscle function. Submaximal $[Ca^{2+}]$ data is reported for OM in the absence of presence of Pi in figure 3, but data for the joint effect of OM and Pi is only reported for maximal $[Ca^{2+}]$. It would greatly strengthen the proposed mechanism if the authors would repeat the Pi experiments reported in figures 4 and 5 at submaximal $[Ca^{2+}]$. This would ensure that cooperative mechanisms prevalent at submaximal $[Ca^{2+}]^2$, whose impacts may be difficult to predict in the context of a drug, do not significantly impact the results.

3. There are important details missing about the development and implementation of the kinetic model shown in figure 6. First, each rate in table 3 should be accompanied with a reference for which the value was obtained from. If the value had to be inferred, then a similar indication should be made in the table with a written or mathematical justification for why or how the parameter was assigned a specific value. Second, the force generating states used to predict the OM/Pi dependencies of the mechanical and kinetic parameters in figures 2, 4, 5 need to be clearly marked in figure 6. Third, at a minimum, the general form of the differential equations underlying the kinetic scheme should be stated in the supplemental information. Ideally, however, the whole system of equations would be written down. Finally, what numerical methods were used to solve the system or what software was used? The inclusion of each of these would significantly add credibility and transparency to the model and following conclusions.

4. As the manuscript mentions (manuscript lines 41-42), it is well known that OM increases the rate of phosphate release (refs 3, 4). However, the kinetic rates found in table 3 do not reflect this; the rate of phosphate release normal XB cycle ($k+4$) is given a value of $25s^{-1}$, while the rate of phosphate release in the OM affected cycle ($k+10$) is only $7s^{-1}$. Please comment on why this is.

5. While it is true that this is the first study to examine OM's effects in situ by combining fast sarcomere-level mechanics and in situ enzyme kinetics, the individual methodologies are well established (refs 5-7). Furthermore, previous studies using different fiber-level techniques have arrived at very similar conclusions as the ones in this study. For instance, OM's mechanism of force enhancement and its potential effect on the muscle's tension cost had both been previously predicted (ref 8). Finally, the manuscript's primary novel contribution, the joint effect of Pi and OM on contractile performance, was not obtained in the most relevant muscle system (cardiac muscle), which limits its usefulness in understanding OM's mechanism. The authors would well be served by making a stronger case for the unique contribution of the data within the context of the substantial literature on OM that already exists.

Minor Comments:

1. In figures 1D and the solid trace in 4B that display the rates of force redevelopment (r_{TD}) as functions of $[OM]$ and $[Pi]$, the points closest to the y-axis are presumably measured under the same experimental conditions: $[OM]=0mM$ and $[Pi]=1mM$. However the furthest left value in 1D is $(3.7\pm 2)s^{-1}$ while the furthest left value in figure 4B is significantly lower, approximately $(2.7\pm 2)s^{-1}$, but increases with $[Pi]$ to a value close to $3.7s^{-1}$. Should this be the case?

2. The methods section explains the experimental apparatus used at the Amsterdam lab to measure ATPase rate. However, it is not clear how the force was measured and recorded during the ATPase experiments. Please clarify this process in the methods section and include any relevant instrument specifications.

3. Please report the temperature at which ATPase rates were measured at the Amsterdam lab.

References

1. Schiaffino, S. & Reggiani, C. Molecular diversity of myofibrillar proteins: gene regulation and functional significance. *Physiol. Rev.* 76, 371–423 (1996).
2. Gordon, A. M., Homsher, E. & Regnier, M. Regulation of contraction in striated muscle. *Physiol. Rev.* 80, 853–924 (2000).
3. Malik, F. I. et al. Cardiac myosin activation: A potential therapeutic approach for systolic heart failure. *Science* (80-.). 331, 1439–1443 (2011).
4. Woody, M. S. et al. Positive cardiac inotrope omecamtiv mecarbil activates muscle despite suppressing the myosin working stroke. *Nat. Commun.* 9, 1–11 (2018).
5. Goldman, Y. Kinetics Of The Actomyosin ATPase In Muscle Fibers. *Annu. Rev. Physiol.* 49, 637–654 (1987).
6. Caremani, M., Melli, L., Dolfi, M., Lombardi, V. & Linari, M. Force and number of myosin motors during muscle shortening and the coupling with the release of the ATP hydrolysis products. *J. Physiol.* 593, 3313–3332 (2015).
7. Percario, V. et al. Mechanical parameters of the molecular motor myosin II determined in permeabilised fibres from slow and fast skeletal muscles of the rabbit. *J. Physiol.* 596, 1243–1257 (2018).
8. Kieu, T. T., Awinda, P. O. & Tanner, B. C. W. Omecamtiv Mecarbil Slows Myosin Kinetics in Skinned Rat Myocardium at Physiological Temperature. *Biophys. J.* 116, 2149–2160 (2019).

Reviewer #2 (Remarks to the Author):

Major Points

The authors explore the effect of phosphate on the action of omecamtiv mecarbil – which surprisingly has a significant effect on muscle mechanics. Overall, this is a thoughtful set of experiments that grow the understanding of how omecamtiv mecarbil works to increase muscle contractility and the energetics of its action. Many will be interested in the conclusions of the paper. However, overall the paper is complex for the reader not already an expert in muscle mechanics. I would suggest the authors simplify the text substantially. The key findings of the paper reviewed in the discussion are lost because of the complexity of the discussion.

Cardiac contraction is more of a twitch (the time of systole ~300 msec is approximately the same as a single turnover of ATP by myosin). Do the results for an isometric contraction differ from what might happen during a short isokinetic contraction? Is there an experiment the authors can do to understand if there is a difference?

In regard to the following sentence and central to the conclusions of the paper:

“The inhibitory effect of OM on force generation is reversed by Pi, which acts as an allosteric competitor that allows the OM-bound no-force-generating motors to release OM and re-enter the force-generating cycle without any change in the rate of ATP hydrolysis.”

How is the alternate model - that of OM stabilizing the lever arm in the pre-powerstroke position, engaging the thin filament, releasing Pi and OM together, and then myosin proceeding to generate force – excluded? Or should it be considered as an alternate model?

I would suggest a table in the text of the paper that describes the various parameters as an easy reference for the reader working through the paper.

Minor Comments

Abstract

The statement below seems to also apply to resting conditions – why the focus on exercise? Suggest you end sentence after the word “patients”.

“This mechanism could underpin an energetically efficient reduction of systolic tension cost in OM-treated patients under exercise, when [Pi] increases with heart-beat frequency.”

Introduction

Misspellings like disfunction (dysfunction) and some other typos and grammatical errors – authors please correct.

Omecamtiv mecarbil not Omecamtiv Mecarbil

This statement - “while the velocity of actin sliding in the in vitro motility assay (IVMA) is reduced 15-17” - is not needed and distracts from the main relevant description of the mechanism of action.

Long sentence – please revise.

“Here we determine the effect of OM on the number and force of the attached myosin motors and on the rate of ATP hydrolysis and their modulation by physiological concentrations of inorganic phosphate (Pi), by applying fast sarcomere level mechanics to Ca²⁺-activated demembranated fibres from rabbit soleus muscle, which contains the slow myosin isoform.”

Suggest use of the word “contractile” rather than “inotropic” in the sentence below. Inotropic generally references an increase in the rate and extent of contractility. OM doesn’t appear to increase the rate of cardiac contractility.

“This effect of Pi on the OM-bound motors represents an energetically efficient inotropic response in patients under OM treatment during exercise and increased workload, when [Pi] increases^{23,24} with the increase of heart beat frequency.”

Results

The authors are applauded for examining the concentration effect relationship of omecamtiv mecarbil in Figure 1. They do focus on the changes at 1 μM . The target range of concentration of omecamtiv mecarbil in humans is 300-750 ng/mL and the free fraction of OM is only 20%. So a plasma concentration of 400 ng/mL translates to a free concentration of 0.2 μM , 5x lower than 1 μM , where the effects are more modest and probably more clinically relevant. The authors should include some text in the results section about the magnitude of the effects in the clinically relevant range of omecamtiv mecarbil concentrations and not just focus on 1 μM .

Unusual to reference to Amsterdam or Florence experiments. The paper’s results usually are described without reference to location.

Generally, the authors should try to simplify the text in the results section if possible. It is difficult to follow – a table of measured parameters would be helpful as mentioned above so that the reader can easily figure out the meaning of the many different variables used.

Discussion

The paper discussion should focus on how these findings contribute to our understanding how OM works in conditions that mimic living, beating cardiac muscle. The relevance of a mechanism of action for omecamtiv mecarbil described previously in the absence of phosphate should be questioned since there is always mM levels of phosphate in working muscle. The authors should compare/contrast their findings with prior findings that did not examine the effects of physiological phosphate and focus on what are the implications to the energetics of cardiac contraction in the presence of omecamtiv mecarbil.

Reviewer #3 (Remarks to the Author):

This contribution from the Lombardi group provides critical insights into the mechanism of Omecamtiv mercarbil (OM), an activator of cardiac myosin that is in Phase 3 clinical trials for heart failure. The data from this laboratory is as always of excellent quality - it combines sarcomere-level mechanics and ATPase measurements in single slow demembrated fibres from rabbit soleus.

A major finding from this paper is the impact of Pi ions on OM-bound myosin heads. In the absence of Pi, OM leads to an increase in ATP consumption due to the inability of OM-bound heads to complete the force-generating power stroke, and thus detaching without completing the powerstroke but having consumed ATP. The addition of physiological levels of ATP reverses the excess ATP consumption and brings it back to the consumption in the absence of drug. This is of great importance since it shows a previously unappreciated effect of OM – its ability to reduce the energetic cost of systolic tension when Pi concentrations increases with heart-beat frequency. Without knowledge this Pi effect, it has been difficult to explain the impact of the drug on the working heart from the experimental data published to date.

While the data is well presented and of very high quality, I have reservations concerning the modeling and its justification. The authors should at least indicate that the model proposed is only one possible model. There is for example no evidence that rebinding of Pi would lead to release of OM as indicated in step 14. In fact there is no basis for suggesting the transition (11) to a state that has a higher affinity for Pi exists. If it is based on other data in the literature, this should be cited. However, it seems simpler to propose that OM increases the occupancy of the state formed by Pi release, allowing exogenous Pi to increase the population of the same state that detaches in the model (green) in the absence of OM. But that OM inhibits its detachment and futile loss of ADP and Pi. In any event, it is important that the authors acknowledge that they have chosen one of the possible models to account for the data since controversy exists as to the states sensitive to Pi rebinding to the myosin head during the powerstroke.

Overall, this is a very strong and well written paper that provides a critical new insight into how a cardiac/slow myosin stimulator can achieve positive results in the clinic. The data is state of the art and the findings will be of wide interest.

Ms NCOMMS-19-32337 (our comments and replies in blue)

We thank the Reviewers for the critiques and suggestions that allowed significant improvement of our paper. While acknowledging the importance and novelty of the paper, they pointed out some weak points, giving us the opportunity to provide a revised version both fully correct and more effective.

Here following are the point-by-point replies to the Reviewers (in blue). All the modifications introduced in the text are highlighted in yellow.

Reply to Reviewer 1

Summary: The authors of the submitted manuscript present data investigating the mechanism by which the systolic heart failure (HF) drug, Omecamtiv Mecarbil (OM), affects the force output and cycling kinetic of rabbit soleus muscle in situ. Mechanical experiments confirmed previously published results that OM increases force at submaximal $[Ca^{2+}]_i$, decreases force near maximal $[Ca^{2+}]_i$, increases calcium sensitivity, and decreases cooperativity. Using a simplified spring-model of the sarcomere, the authors show that the OM-induced reduction in force at max $[Ca^{2+}]_i$ can be attributed to the reduction in the average strain experienced by each attached motor, as opposed to a reduction in the number of bound motors. At submaximal $[Ca^{2+}]_i$, OM-induced force increases were attributed to an increase in the number of bound motors, which supports previous findings. Interestingly, the addition of physiological concentrations of inorganic phosphate (Pi) significantly mitigates the OM-induced force reduction at maximal $[Ca^{2+}]_i$. The addition of Pi also had significant dose dependent effects on the rate of force redevelopment and the average motor strain. As a second source of kinetic insight, the authors measured the overall ATPase rate (k_{cat}) in isometrically contracting muscle to find that OM did not significantly impact the Pi dependence of k_{cat} but did reduce the muscle's tension cost. As the authors mention, this finding is specific to the context of high load contraction. To explain the observed effects of OM and Pi, the authors extend a previously published kinetic model of muscle myosin. The main feature of the model is the addition of an OM pathway that inhibits the power stroke and introduces an isomerization step following $P \rightarrow i$ where the new state has an increased Pi affinity. The new state allows for competitive interactions between OM and $P \rightarrow i$ in which $P \rightarrow i$ can allosterically displace OM, thus allowing the motor to reenter the usual force producing ATPase cycle. Finally, the authors suggest that their experimental findings and model predictions may identify an energy-efficient response to OM during exercise when intercellular Pi increases within the heart.

Major Comments:

1. While it is understandable from the experimental perspective why OM's effects were studied in soleus skeletal muscle instead of cardiac muscle, the authors translate results between the two systems without much discussion of potential limitations and caveats. Such translation may be appropriate for single molecule results as the dominant myosin isoform in soleus and cardiac muscle are so similar (ref 1) but this is harder to justify at the fiber level where other important sarcomeric proteins differ in isoform and functional role. Furthermore, considering that the vast majority of fiber level studies on OM are in cardiac muscle and that little has been previously published on OM's effect on soleus skeletal muscle, a detailed limitations section and/or additional experimental evidence is needed to contextualize/justify the translation from soleus to cardiac muscle.

We share the reviewer concern about the different functional role of other sarcomere proteins on skeletal and cardiac muscle and we took it into consideration in the revised text. In particular it is mandatory to take into account the contribution of other mechanically relevant proteins that may weaken the analysis of half-sarcomere compliance in terms of the simple model (defined by eqn (1)) made by the array of myosin motors in series with the equivalent filament compliance. For this we have moved the half-sarcomere compliance analysis from SI to Methods, to make more explicit the conditions under which eqn (1) can be applied to estimate the number of attached motors.

It was our first aim to exploit sarcomere-level mechanics in cardiac myocytes to define in situ the action mechanism of OM on β -cardiac myosin and reveal the effect of physiological levels of Pi. The reasons that make the skinned fibre from rabbit soleus muscle the preparation of election for this study are now explained in more detail in the revised text. The intact trabecula dissected from the ventricle of the rat heart, the only cardiac preparation on which our sarcomere-level mechanics has been successfully exploited (Caremani et al., 2016; Pinzauti et al., 2018), does not suit the requirements of the present investigation because (i) only skinned myocytes allow the required manipulation of $[Ca^{2+}]$ and $[Pi]$ and (ii), unlike rabbit soleus that has almost 100% of β /slow MHC isoform (Percario et al., 2018) as the human ventricle, in the rat trabecula only ~20% of the myosin is the β /slow MHC isoform and ~80% is α MHC isoform (Pinzauti et al, 2018).

2. In the introduction (manuscript lines 63-69) the authors use the fact that intracellular $[Ca^{2+}]$ remains below saturating levels during left ventricular systole in their proposed mechanism explaining how OM and Pi jointly impact muscle function. Submaximal $[Ca^{2+}]$ data is reported for OM in the absence of presence of Pi in figure 3, but data for the joint effect of OM and Pi is only reported for maximal $[Ca^{2+}]$. It would greatly strengthen the proposed mechanism if the authors would repeat the Pi experiments reported in figures 4 and 5 at submaximal $[Ca^{2+}]$. This would ensure that cooperative mechanisms prevalent at submaximal $[Ca^{2+}]^2$, whose impacts may be difficult to predict in the context of a drug, do not significantly impact the results.

This is a crucial argument that requires a clarification on the reasons that prevented us to extend the analysis of the Pi effect to submaximal Ca^{2+} . The clarification is introduced in the revised text by moving the half-sarcomere compliance analysis and the related figure (now Fig. 7) from SI to Methods. Independent of the presence/absence of OM, Pi experiments at submaximal Ca^{2+} cannot provide a reliable estimate of the number of attached motors, because the stiffness of the motor array would be too low (<20 kPa/nm, Fig. 3B) and the half-sarcomere compliance analysis would be complicated by the significant contribution of the additional non-cross-bridge elasticity in parallel with the motor array (see Methods, Fig. 7D and integrated Discussion). To demonstrate the solidity of our analysis and conclusions in this respect, we have attached to this reply a version of Figure 7C and D in which the mean points are replaced by pooled data obtained from the T1 relations (Fig. R1).

There are important details missing about the development and implementation of the kinetic model shown in figure 6. First, each rate in table 3 should be accompanied with a reference for which the value was obtained from. If the value had to be inferred, then a similar indication should be made in the table with a written or mathematical justification for why or how the parameter was assigned a specific value.

We apologise for having missed to give the kinetic model details and especially for the scarce accuracy used in constraining the kinetic steps that were not crucial for the demonstration of the new finding of the isomerisation on the OM-motors with change in affinity followed by the Pi-OM exchange. The model has been revised considering all the kinetic constraints from the literature (see Table 3) and the detailed justification of each kinetic step (see last column in Table 3) and the text has been integrated accordingly. Noteworthy, the correction of the model with all the constrains from the literature does not change the results of the simulation and the important conclusion of the paper that the isomerisation of the OM bound motors followed by Pi-OM exchange is the only kinetic feature able to explain the combined OM-Pi effects. In this respect see Figs. R2, R3 and R4 enclosed here, where the data in the corresponding Figs. 1, 4 and 5 of the revised manuscript (symbols data, black lines simulation) are superimposed on the old simulation (red lines). We thank Reviewer 1 for promoting the revision that makes the kinetic simulation correct and consistent with previous data in the literature.

Second, the force generating states used to predict the OM/Pi dependencies of the mechanical and kinetic parameters in figures 2, 4, 5 need to be clearly marked in figure 6.

Fig. 6 legend integrated accordingly.

Third, at a minimum, the general form of the differential equations underlying the kinetic scheme should be stated in the supplemental information. Ideally, however, the whole system of equations would be written down.

The system of equations is now reported in SI.

Finally, what numerical methods were used to solve the system or what software was used? The inclusion of each of these would significantly add credibility and transparency to the model and following conclusions.

Done as requested.

4. As the manuscript mentions (manuscript lines 41-42), it is well known that OM increases the rate of phosphate release (refs 3, 4). However, the kinetic rates found in table 3 do not reflect this; the rate of phosphate release normal XB cycle ($k+4$) is given a value of $25s^{-1}$, while the rate of phosphate release in the OM affected cycle ($k+10$) is only $7s^{-1}$. Please comment on why this is.

We agree with the Reviewer about the unjustified assumption for this kinetic parameter. As admitted in the preliminary general reply to the Editor and Reviewers, our effort was mainly dedicated to try and discard any alternative hypothesis to the relevant finding in this work, that is the isomerisation of the OM-bound state with change in Pi affinity. We thank the Reviewer for triggering the revisions that make the kinetic scheme coherent with data in the literature (listed in Table 3). In particular $k+4$ and $k+10$ are now chosen according to Liu et al, 2015 and Woody et al, 2018.

5. While it is true that this is the first study to examine OM's effects in situ by combining fast sarcomere-level mechanics and in situ enzyme kinetics, the individual methodologies are well established (refs 5-7).

References to the first papers reporting the individual methodologies (Linari et al., 2007; Percario et al., 2018, for the mechanics; Glyn & Sleep 1985; Potma et al., 1994 for the ATPase) are already in the list.

Furthermore, previous studies using different fiber-level techniques have arrived at very similar conclusions as the ones in this study. For instance, OM's mechanism of force enhancement and its potential effect on the muscle's tension cost had both been previously predicted (ref 8).

Reference to Kieu et al. 2019 has been added.

Finally, the manuscript's primary novel contribution, the joint effect of Pi and OM on contractile performance, was not obtained in the most relevant muscle system (cardiac muscle), which limits its usefulness in understanding OM's mechanism. The authors would well be served by making a stronger case for the unique contribution of the data within the context of the substantial literature on OM that already exists.

As acknowledged by all the Reviewers, our finding of the effect of physiological concentrations of Pi on the force of OM-bound motors is an important contribution that questions all the previous work on the inotropic mechanism of OM on β /slow MHC isoform in the absence of Pi. Its translation to cardiac muscle is now discussed in more detail (see also the detailed argument in the introduction of this reply to the Editor and Reviewers).

Minor Comments:

1. In figures 1D and the solid trace in 4B that display the rates of force redevelopment (rTD) as functions of

[OM] and [Pi], the points closest to the y-axis are presumably measured under the same experimental conditions: [OM]=0mM and [Pi]=1mM. However the furthest left value in 1D is $(3.7\pm 2)s^{-1}$ while the furthest left value in figure 4B is significantly lower, approximately $(2.7\pm 2)s^{-1}$, but increases with [Pi] to a value close to $3.7s^{-1}$. Should this be the case?

We thank the Reviewer for noting the contradiction. r_{TD} plotted in Fig. 1 D has been calculated as the reciprocal of the half-time of force development following a temperature jump, as defined in the text while r_{TD} plotted in Fig. 4B has been calculated as the reciprocal of the time constant of force development (used in a preliminary version of the analysis) and thus its value is significantly smaller. We apologize for this and now we have given a consistent estimate to r_{TD} , reporting also in Fig 4B its value as the reciprocal of half-time.

2. The methods section explains the experimental apparatus used at the Amsterdam lab to measure ATPase rate. However, it is not clear how the force was measured and recorded during the ATPase experiments. Please clarify this process in the methods section and include any relevant instrument specifications.

Information about the force transducer used for ATPase measurements added.

3. Please report the temperature at which ATPase rates were measured at the Amsterdam lab.

Temperature of Amsterdam experiments (12 °C) added.

References

1. Schiaffino, S. & Reggiani, C. Molecular diversity of myofibrillar proteins: gene regulation and functional significance. *Physiol. Rev.* 76, 371–423 (1996).
2. Gordon, A. M., Homsher, E. & Regnier, M. Regulation of contraction in striated muscle. *Physiol. Rev.* 80, 853–924 (2000).
3. Malik, F. I. et al. Cardiac myosin activation: A potential therapeutic approach for systolic heart failure. *Science* (80-.). 331, 1439–1443 (2011).
4. Woody, M. S. et al. Positive cardiac inotrope omecamtiv mecarbil activates muscle despite suppressing the myosin working stroke. *Nat. Commun.* 9, 1–11 (2018).
5. Goldman, Y. Kinetics Of The Actomyosin ATPase In Muscle Fibers. *Annu. Rev. Physiol.* 49, 637–654 (1987).
6. Caremani, M., Melli, L., Dolfi, M., Lombardi, V. & Linari, M. Force and number of myosin motors during muscle shortening and the coupling with the release of the ATP hydrolysis products. *J. Physiol.* 593, 3313–3332 (2015).
7. Percario, V. et al. Mechanical parameters of the molecular motor myosin II determined in permeabilised fibres from slow and fast skeletal muscles of the rabbit. *J. Physiol.* 596, 1243–1257 (2018).
8. Kieu, T. T., Awinda, P. O. & Tanner, B. C. W. Omecamtiv Mecarbil Slows Myosin Kinetics in Skinned Rat Myocardium at Physiological Temperature. *Biophys. J.* 116, 2149–2160 (2019).

Reply to Reviewer 2

Major Points

The authors explore the effect of phosphate on the action of omecamtiv mecarbil – which surprisingly has a significant effect on muscle mechanics. Overall, this is a thoughtful set of experiments that grow the understanding of how omecamtiv mecarbil works to increase muscle contractility and the energetics of its action. Many will be interested in the conclusions of the paper. However, overall the paper is complex for the reader not already an expert in muscle mechanics. I would suggest the authors simplify the text substantially. The key findings of the paper reviewed in the discussion are lost because of the complexity of the discussion.

We revised the Introduction and the Discussion to give a more efficient description of the unicity of our approach for the *in situ* definition of OM action on β /slow MHC isoform of the myosin motor and of the relevance of the finding of the inotropic effect of physiological concentrations of Pi on the OM-bound myosin motor.

Cardiac contraction is more of a twitch (the time of systole ~300 msec is approximately the same as a single turnover of ATP by myosin). Do the results for an isometric contraction differ from what might happen during a short isokinetic contraction? Is there an experiment the authors can do to understand if there is a difference?

To answer this question requires a completely different set of mechanical experiments planned on a different preparation as the intact trabecula of rat heart. On the other hand, those new experiments, when possible, can only be conceived after this work, in which, with the sarcomere-level mechanics on skinned fibres, the effects of Ca and Pi modulation on the number and force of myosin motors in the absence and presence of OM have been defined.

In regard to the following sentence and central to the conclusions of the paper:

“The inhibitory effect of OM on force generation is reversed by Pi, which acts as an allosteric competitor that allows the OM-bound no-force-generating motors to release OM and re-enter the force-generating cycle without any change in the rate of ATP hydrolysis.”

How is the alternate model - that of OM stabilizing the lever arm in the pre-powerstroke position, engaging the thin filament, releasing Pi and OM together, and then myosin proceeding to generate force – excluded? Or should it be considered as an alternate model?

The alternative model proposed by the Reviewer is one of the many preliminarily tested and has been discarded because it implies (i) the reduction of the force, also in the presence of OM, with the increase in Pi, which is in contradiction with the results in this work, and (ii) the release of OM from the attached motor that could then detach only by binding of ATP, which is contradicted by Woody et al. (2018) finding that OM-bound motors detach without implying ATP binding.

I would suggest a table in the text of the paper that describes the various parameters as an easy reference for the reader working through the paper.

Done as requested

Minor Comments

Abstract

The statement below seems to also apply to resting conditions – why the focus on exercise? Suggest you end sentence after the word “patients”. “This mechanism could underpin an energetically efficient reduction of systolic tension cost in OM-treated patients under exercise, when [Pi] increases with heart-beat frequency.”

Sentence reworded here and throughout the text removing the mention to exercise.

Introduction

Misspellings like disfunction (dysfunction) and some other typos and grammatical errors – authors please correct.

Done

Omecamtiv mecarbil not Omecamtiv Mecarbil

Done

This statement - “while the velocity of actin sliding in the in vitro motility assay (IVMA) is reduced 15-17” - is not needed and distracts from the main relevant description of the mechanism of action.

Long sentence – please revise.

“Here we determine the effect of OM on the number and force of the attached myosin motors and on the rate of ATP hydrolysis and their modulation by physiological concentrations of inorganic phosphate (Pi), by applying fast sarcomere level mechanics to Ca²⁺-activated demembranated fibres from rabbit soleus muscle, which contains the slow myosin isoform.”

Sentence splits in two

Suggest use of the word “contractile” rather than “inotropic” in the sentence below. Inotropic generally references an increase in the rate and extent of contractility. OM doesn’t appear to increase the rate of cardiac contractility. “This effect of Pi on the OM-bound motors represents an energetically efficient inotropic response in patients under OM treatment during exercise and increased workload, when [Pi] increases^{23,24} with the increase of heart beat frequency.”

The adjective related to effects on the heart rate is “chronotropic”, we would like to keep “inotropic” for the adjective specifically defining the effects on the force

Results

The authors are applauded for examining the concentration effect relationship of omecamtiv mecarbil in Figure 1. They do focus on the changes at 1 uM. The target range of concentration of omecamtiv mecarbil in humans is 300-750 ng/mL and the free fraction of OM is only 20%. So a plasma concentration of 400 ng/mL translates to a free concentration of 0.2 uM, 5x lower than 1 uM, where the effects are more modest and probably more clinically relevant. The authors should include some text in the results section about the magnitude of the effects in the clinically relevant range of omecamtiv mecarbil concentrations and not just focus on 1 uM.

The reviewer’s suggestion points at one of the several important experiments conceivable after the discovery of the combined OM-Pi effects in this work, namely the effect of Pi at lower, more clinically relevant, concentrations of OM. This point is explicitly discussed in the revised version. Another crucial point to investigate, even if at lower resolution than the fast half-sarcomere mechanics exploited here, will be the Pi effect on OM-bound motors at subsaturating [Ca²⁺] in demembranated trabeculae of rat heart in both isometric and isotonic conditions.

Unusual to reference to Amsterdam or Florence experiments. The paper's results usually are described without reference to location.

Reference to location removed

Generally, the authors should try to simplify the text in the results section if possible. It is difficult to follow – a table of measured parameters would be helpful as mentioned above so that the reader can easily figure out the meaning of the many different variables used.

We introduced the list of abbreviations used for the parameters.

Discussion

The paper discussion should focus on how these findings contribute to our understanding how OM works in conditions that mimic living, beating cardiac muscle. The relevance of a mechanism of action for omecamtiv mecarbil described previously in the absence of phosphate should be questioned since there is always mM levels of phosphate in working muscle. The authors should compare/contrast their findings with prior findings that did not examine the effects of physiological phosphate and focus on what are the implications to the energetics of cardiac contraction in the presence of omecamtiv mecarbil.

We thank the Reviewer for this suggestion that we have literally introduced in the Discussion to highlight the clinically most relevant conclusion of the paper.

Reply to Reviewer 3

This contribution from the Lombardi group provides critical insights into the mechanism of Omecamtiv mecarbil (OM), an activator of cardiac myosin that is in Phase 3 clinical trials for heart failure. The data from this laboratory is as always of excellent quality - it combines sarcomere-level mechanics and ATPase measurements in single slow demembrated fibres from rabbit soleus.

A major finding from this paper is the impact of Pi ions on OM-bound myosin heads. In the absence of Pi, OM leads to an increase in ATP consumption due to the inability of OM-bound heads to complete the force-generating power stroke, and thus detaching without completing the powerstroke but having consumed ATP. The addition of physiological levels of Pi reverses the excess ATP consumption and brings it back to the consumption in the absence of drug. This is of great importance since it shows a previously unappreciated effect of OM – its ability to reduce the energetic cost of systolic tension when Pi concentrations increases with heart-beat frequency. Without knowledge this Pi effect, it has been difficult to explain the impact of the drug on the working heart from the experimental data published to date. While the data is well presented and of very high quality, I have reservations concerning the modeling and its justification. The authors should at least indicate that the model proposed is only one possible model. There is for example no evidence that rebinding of Pi would lead to release of OM as indicated in step 14. In fact there is no basis for suggesting the transition (11) to a state that has a higher affinity for Pi exists. If it is based on other data in the literature, this should be cited. However, it seems simpler to propose that OM increases the occupancy of the state formed by Pi release, allowing exogenous Pi to increase the population of the same state that detaches in the model (green) in the absence of OM. But that OM inhibits its detachment and futile loss of ADP and Pi. In any event, it is important that the authors acknowledge that they have chosen one of the possible models to account for the data since controversy exists as to the states sensitive to Pi rebinding to the myosin head during the powerstroke.

We took into careful account both the argument of the Reviewer about the question that ours is not the unique scheme able to explain the results and the suggestion of the simpler scheme based on the allosteric competition between OM and Pi. This latter scheme was the first we used in our trials to simulate the results and was discarded because it failed to reproduce (i) the OM-dependent drop in the rate of force development, (ii) the recovery of the average force per motor by Pi in the presence of OM. We revised the text to give a more detailed explanation of the unicity of our kinetic scheme for interpreting the results. As far as the relation between Pi release and the working stroke, our model is based on the experimental evidence, from the many papers of others and ours, that the force generating step may occur also before Pi release (Dantzig et al., 1992; Millar & Homsher., 1992; Caremani et al., 2008,2013, 2015; Woody et al., 2018).

Overall, this is a very strong and well written paper that provides a critical new insight into how a cardiac/slow myosin stimulator can achieve positive results in the clinic. The data is state of the art and the findings will be of wide interest.

Fig. R1. A. Relations of Y_0 versus T_0 at different pCa determined in control (filled circles) and in the presence of 1 μ M OM (open circles). The points are the pooled data that contribute to the mean data plotted in text Figure 7C. The lines are linear regressions on the pooled data for forces > 30 kPa in control (continuous) and in the presence of OM (dashed). B. Relations of half-sarcomere strain (Y_0) versus the stiffness of the myosin motors at different pCa (e_0) in control (filled circles) and in the presence of 1 μ M OM (open circles). The points are the pooled data that contribute to the mean data plotted in text Figure 7D. The lines are linear regressions on the pooled data for stiffness > 20 kPa/nm in control (continuous) and in the presence of OM (dashed).

Fig. R2. From Fig. 1 of the revised version of the manuscript (black symbols and lines). Red lines are the results from the simulation reported in the first version of the manuscript.

Fig. R3. From Fig. 4 of the revised version of the manuscript (black symbols and lines). Red lines are the results from the simulation reported in the first version of the manuscript

Fig. R4. From Fig. 5 of the revised version of the manuscript (black symbols and lines). Red lines are the results from the simulation reported in the first version of the manuscript.

Reviewers' comments:

Reviewer #1 (Remarks to the Author):

The authors have resubmitted a greatly improved version of their manuscript. I only have a few additional comments on the rebuttal responses and associated manuscript updates.

Major comments:

1. The authors justify their choice to use permeabilized rabbit soleus muscle preparations over a permeabilized cardiac muscle system with (i) the lab's previous experiences with skinned rabbit soleus and (ii) the want to have a homogeneous ensemble of beta-MHC. In response to the first point, what prevents the authors from using skinned rat trabeculae? There are many examples of other studies successfully using skinned rat trabeculae to study OM.

The reasoning behind the second argument is understandable as human cardiac MHC, as the authors mention, is predominantly of the beta isoform. However, it seems that it would still be preferable to study the joint effects of [OM] and [Pi] in the rat cardiac system. This data would be more readily compared to previous OM studies and more translatable to other cardiac systems despite the differences in MHC isoform expression. Additionally, the work in (Pinzauti et al. 2018) shows that a similar model of sarcomere compliance and strain can be applied to rat trabeculae; comparing table 2 in (Pinzauti et al. 2018) to table 2 in (Percario et al. 2018) further reveals that the necessary parameters for skinned rat trabeculae could be determined. Were there other more substantial limitations to using skinned rat cardiac trabeculae that were not mentioned in the manuscript and rebuttal?

2. The addition of Table 3 greatly improved the clarity and presentation of the proposed kinetic model. It would further help if the authors would add uncertainties to the parameters that were "adjusted" or "set" to indicate how sensitive the model fit is.

3. The authors should include some goodness-of-fit metric to quantify how well the model fits the experimental data.

Minor Comments:

1. In the system of equations variables for [OM], [ATP], and [ADP] lack the square brackets. This makes them difficult to distinguish from the other variables in the equations.

References:

Pinzauti, F., I. Pertici, M. Reconditi, T. Narayanan, G.J.M. Stienen, G. Piazzesi, V. Lombardi, M. Linari, and M. Caremani. 2018. The force and stiffness of myosin motors in the isometric twitch of a cardiac trabecula and the effect of the extracellular calcium concentration. *J. Physiol.* 596:2581–2596. doi:10.1113/JP275579.

Percario, V., S. Boncompagni, F. Protasi, I. Pertici, F. Pinzauti, and M. Caremani. 2018. Mechanical parameters of the molecular motor myosin II determined in permeabilised fibres from slow and fast skeletal muscles of the rabbit. *J. Physiol.* 596:1243–1257. doi:10.1113/JP275404.

Reviewer #2 (Remarks to the Author):

The authors have addressed my comments in their revisions and I thank them for their consideration of these comments. The paper is strengthened by their revisions.

I have only the following minor suggestions for the Introduction:

1) After the first sentence in the introduction, I would include a sentence about heart failure with reduced ejection fraction. While this is an acquired cardiomyopathy, increasing cardiac contractility may also play an important role in treatment.

2) In the third sentence of the introduction - omecamtiv should not be capitalized.

Reviewer #3 (Remarks to the Author):

The authors have provided a satisfactory answer to my questions.

The paper is adequate for publication.

Reply to reviewers (our comments and replies in blue)

Reviewer #1

The authors have resubmitted a greatly improved version of their manuscript. I only have a few additional comments on the rebuttal responses and associated manuscript updates.

Major comments:

1. The authors justify their choice to use permeabilized rabbit soleus muscle preparations over a permeabilized cardiac muscle system with (i) the lab's previous experiences with skinned rabbit soleus and (ii) the want to have a homogeneous ensemble of beta-MHC. In response to the first point, what prevents the authors from using skinned rat trabeculae? There are many examples of other studies successfully using skinned rat trabeculae to study OM.

None of the previous studies on skinned trabeculae, included our attempts, attained the sarcomere-level resolution and the time resolution required in this work to define the stoichiometry of OM and Pi effects on the number of attached motors, which made possible to reveal the effect of physiological concentrations of Pi on the efficiency of OM-treated fibres.

The reasoning behind the second argument is understandable as human cardiac MHC, as the authors mention, is predominantly of the beta isoform. However, it seems that it would still be preferable to study the joint effects of [OM] and [Pi] in the rat cardiac system. This data would be more readily compared to previous OM studies and more translatable to other cardiac systems despite the differences in MHC isoform expression. Additionally, the work in (Pinzauti et al. 2018) shows that a similar model of sarcomere compliance and strain can be applied to rat trabeculae; comparing table 2 in (Pinzauti et al. 2018) to table 2 in (Percario et al. 2018) further reveals that the necessary parameters for skinned rat trabeculae could be determined. Were there other more substantial limitations to using skinned rat cardiac trabeculae that were not mentioned in the manuscript and rebuttal?

In the revised version of the paper, submitted on January 13th, we already gave (at Page 2, 2nd para, copied below) the reasons why we chose skinned rabbit soleus fibres instead of skinned rat trabeculae:

“The slow skeletal muscle of the rabbit has been chosen instead of the heart because their myosin isoforms exhibit similar affinity for OM⁸, while the required nanometer-microsecond resolution of sarcomere-level mechanics can only be achieved in demembranated myocytes from skeletal muscle^{18,19}. Sarcomere-level mechanics has been recently successfully exploited in intact trabeculae dissected from the ventricle of the rat heart^{20,21}, but that preparation does not suit the present investigation because (i) only skinned myocytes allow the required manipulation of [Ca²⁺] and [Pi] and (ii), unlike rabbit soleus that has almost 100% of β /slow MHC isoform¹⁹, in the rat trabecula only ~20% of the myosin is β /slow MHC isoform and the remaining ~80% is α MHC isoform²¹”.

When the Reviewer says “...comparing table 2 in (Pinzauti et al. 2018) to table 2 in (Percario et al. 2018) further reveals that the necessary parameters for skinned rat trabeculae could be determined”, he/she appears not to consider the fact that in Pinzauti et al 2018 (Ref 21) the experiments were carried out on electrically paced intact trabeculae, because only in the intact

preparation the resolution of sarcomere level mechanics that allows the estimate of the stiffness, and thus of the number of attached motors, can be achieved. Unfortunately, (i) the intact preparation cannot be used to study the dependence of these parameters on myofibrillar $[Ca^{2+}]$ and $[Pi]$ and (ii) the skinned trabeculae that should be used in order to apply the protocols that require manipulation of $[Ca^{2+}]$ and $[Pi]$ do not allow the required sarcomere-level resolution. Under these conditions it is justified from methodological point of view the use of skinned slow muscle fibres from rabbit, which have the same β /slow isoform as the trabeculae of large mammals.

Following the first review we thought that this was also the opinion of the Reviewer when he /she says (first line of Main Comments): *“1. While it is understandable from the experimental perspective why OM’s effects were studied in soleus skeletal muscle instead of cardiac muscle...”*

2. The addition of Table 3 greatly improved the clarity and presentation of the proposed kinetic model. It would further help if the authors would add uncertainties to the parameters that were “adjusted” or “set” to indicate how sensitive the model fit is.

We thank the Reviewer for the request of clarification on the procedure followed to assign a specific value to the kinetic parameters concerning the effects of OM and Pi on the force, rate of force development, stiffness of the motor array and ATPase rate (in total eight relations). The predictions of the simulation for the eight relations are optimised by an iterative procedure in which the values of the rate constants controlling the kinetics of the relevant steps (8-12 and 14) are changed one at a time. As explained in detail in Methods the residual errors of each trial have been calculated for selecting the value of each parameter with the minimum residual. An example is given in Figure S2, where in the left column the simulated relations obtained by changing k_{+14} are superimposed to the experimental relations and in the right column the corresponding residuals are shown. Since for any given kinetic parameter the sensitivity of the test in showing a minimum mean residual and its dependence on the value of the parameter vary depending on the relation, to select the value of each parameter we calculated the global mean residuals by averaging the mean residuals over the eight relevant relations (see Figure S3). The plots in this Figure make explicit the sensitivity of the simulation to each kinetic parameter providing the range of values admitted following the optimisation procedure. The Reviewer request of defining a rigorous iterative method to optimise the minimisation of residual errors between simulated and experimental relations, let us discover that for step 12 the minimum global mean residual is attained with $k_{+12} = 1.5/s$ (see Figure S3J) instead of $2/s$ as reported in the original paper. The simulated relations in all the involved Figures (1C, 1D, 2D, 4A-D, 5B-E) have been replotted, even if none of the graphs shows appreciable changes for a reduction in k_{+12} by 25%.

3. The authors should include some goodness-of-fit metric to quantify how well the model fits the experimental data.

Done by calculating, for each kinetic parameter, the mean residuals for each of the eight relevant relations and then the global mean residual averaged over the eight relations.

Minor Comments:

1. In the system of equations variables for $[OM]$, $[ATP]$, and $[ADP]$ lack the square brackets. This makes them difficult to distinguish from the other variables in the equations.

Square brackets added to OM, ATP, ADP and Pi

Reviewer #2

The authors have addressed my comments in their revisions and I thank them for their consideration of these comments. The paper is strengthened by their revisions.

I have only the following minor suggestions for the Introduction:

1) After the first sentence in the introduction, I would include a sentence about heart failure with reduced ejection fraction. While this is an acquired cardiomyopathy, increasing cardiac contractility may also play an important role in treatment.

The text has been integrated as requested

2) In the third sentence of the introduction - omecamtiv should not be capitalized.

Done (also in the title)

Reviewer #3 (Remarks to the Author):

The authors have provided a satisfactory answer to my questions.
The paper is adequate for publication.